# BACKDOOR UNLEARNING BY LINEAR TASK DECOMPOSITION

## ABSTRACT

Foundation models have revolutionized computer vision by enabling broad generalization across diverse tasks. Yet, they remain highly susceptible to adversarial perturbations and targeted backdoor attacks. Mitigating such vulnerabilities remains an open challenge, especially given that the large scale of the models prohibits retraining to ensure safety. Existing backdoor removal approaches rely on costly fine-tuning to override the harmful behavior, and can often degrade performance on other unrelated tasks. This raises the question of whether backdoors can be unlearned without compromising the general capabilities of the models. In this work, we study how backdoors are encoded in the model weight space and find that they are *disentangled* from other benign tasks. Specifically, this separation enables the isolation and erasure of the backdoor's influence on the model's weights with minimal impact on clean performance. Building on this insight, we introduce a simple unlearning method that leverages such disentanglement. Through extensive experiments with CLIP-based models and common adversarial triggers, we show that, given the knowledge of the attack, our method achieves approximately perfect unlearning, while retaining on average 96% of clean accuracy. Additionally, we demonstrate that even when the type of attack is unknown, our method successfully unlearns backdoors by proper estimation using reverse-engineered triggers. Overall, our method consistently yields better approximate unlearning and clean accuracy tradeoffs when compared to present state-of-the-art defenses.

## 1 INTRODUCTION

Foundation models have become a cornerstone of modern deep learning, offering broad generalization across a wide range of tasks through large-scale pre-training (Radford et al., 2021; Jia et al., 2021). Among them, vision-language models like CLIP (Radford et al., 2021) play a fundamental role. They not only demonstrate remarkable robustness to distribution shifts and *zero-shot* performance on out-of-distribution benchmarks (Wortsman et al., 2022b), but their vision encoders also serve as a key component in other state-of-the-art large vision-language models (LVLMs), such as LLaVA, where they are paired with a large language model (LLM).

However, the very success and widespread integration of these models make them a prime target for security threats, most notably *backdoor attacks* (Carlini & Terzis, 2021; Bansal et al., 2023) – a class of threats that compromise model integrity even after training is complete. In a backdoor attack (Gu et al., 2017), an adversary poisons a small portion of the training data by embedding a fixed trigger pattern into inputs and mislabeling them to a target class. The resulting model appears to perform well on clean inputs but systematically misclassifies any input containing the trigger – effectively granting the adversary precise control over model predictions. Such vulnerabilities pose a serious risk in safety-critical applications, including autonomous driving and medical diagnostics (Du et al., 2024; Hanif et al., 2024).

Current defenses for CLIP largely fall into two categories: *(i)* retraining the model from scratch using modified loss functions designed to resist backdoors, or *(ii)* fine-tuning on clean data to override the malicious behavior (Bansal et al., 2023; Yang et al., 2024b; Goel et al., 2022a). However, full retraining is prohibitively expensive at scale, while fine-tuning – though cheaper – frequently induces *catastrophic forgetting*(French, 1999), whereby the pre-trained knowledge is erased. Furthermore, recent studies show that fine-tuning strategies struggle against more sophisticated attacks (Liang et al., 2024).

An alternative line of work is *machine unlearning* (Cao & Yang, 2015), which seeks to selectively remove (or *forget*) specific learned behaviors post-hoc, avoiding full retraining. Currently, the application of unlearning methods to targeted backdoor removal remains limited. For instance, prominent unlearning algorithms such as *gradient ascent* and its variants have been shown to fall short when applied to backdoor removal in small-scale settings (Pawelczyk et al., 2024), and their effectiveness in large-scale foundation models remains an open question.

In this paper, we introduce an efficient, post-hoc method for removing backdoors from vision-language foundation models while preserving their clean capabilities. Our approach builds on recent advances in model editing in weight space (Frankle et al., 2020; Izmailov et al., 2018; Wortsman et al., 2021; 2022a; Rame et al., 2022; Ainsworth et al., 2022; Ilharco et al., 2022b;a). In particular, Ilharco et al. (2022a) introduced the concept of *task vector*: the element-wise difference between the weights of a fine-tuned model and its pre-trained initialization. Task vectors provide a means to encode learned tasks as directions in weight space. They can be added to a model to inject functionality, subtracted to unlearn specific tasks, or combined to compose multi-task models. These manipulations are enabled by the *disentanglement* of tasks in the weight space of pre-trained models, as recently formalized by Ortiz-Jimenez et al. (2024).

Motivated by these insights, we investigate how backdoors are encoded in the weight space of CLIP-based models. We show that the weights can be linearly decomposed into clean and triggered components, effectively disentangling the malicious behavior from the model's benign capabilities. This disentanglement allows us to isolate the backdoor's influence by exploiting task arithmetic. This is achieved by fine-tuning the model on a small set of triggered examples to compute a "trigger vector". This vector isolates the malicious behavior and can thus be subtracted – via task negation – to surgically remove the backdoor while preserving clean model performance (illustrated in Figure 1).

Our main contributions are:

- We demonstrate that backdoors in CLIP-based transformer models are disentangled from clean knowledge in weight space, enabling targeted removal via linear operations without encountering catastrophic forgetting of non-adversarial knowledge.

- We introduce TBAR (**T**rigger removal by **B**ackdoor **AR**ithmetic), a lightweight approach for approximate backdoor unlearning via weight-space task negation. On *(i)* image-classification backdoor benchmarks and *(ii)* large-scale image-caption settings where the trigger is known, TBAR unlearns 99% of the backdoor while retaining on average 96% clean accuracy. In particular, in the latter case, it outperforms state-of-the-art clean-data fine-tuning defenses while using less than 2% of the data requirements.

- We extend TBAR to operate in large-scale settings in an attack-agnostic scenario by pairing it with reverse-engineered proxy triggers. Our method successfully sanitizes infected models, outperforming state-of-the-art defenses while preserving over 90% clean accuracy.

## 2 PROBLEM SETUP

**Threat model** The adversary has full white-box access to a pre-trained model and fine-tuning data to be used to backdoor the model. The attack is conducted by injecting a small poisoned subset into a larger training dataset. The resulting backdoored model is released publicly and intended for downstream use by unaware users. Unless otherwise specified we consider the attack successful if the triggered examples are predicted as a targeted label.

**Defender assumptions** The defender's goal is to remove the backdoor (i.e., reduce attack success rate to zero) while preserving the model's performance on clean data. The defender has full access to model weights. We consider two distinct practical scenarios:

- **Trigger-known:** The defender is given a small forget set containing the true trigger, reflecting a common assumption in the context of backdoor defenses within unlearning studies, where an attack has been identified and its characteristics are known.

- **Trigger-unknown:** The defender does not know the true trigger but has access to a small set of clean data.

## 3 BACKGROUND

**Notation**   Let a neural network be a parameterized function $f : \mathcal{X} \times \Theta \to \mathcal{Y}$ with inputs $x \in \mathcal{X}$ and weights $\boldsymbol{\theta} \in \Theta$. A task $k \in [K]$ is identified as a triplet $(\mathcal{D}_k, \mu_k, f_k^\star)$ with domain $\mathcal{D}_k \subseteq \mathcal{X}$, input distribution $\mu_k$ ($\mathrm{supp}(\mu_k) = \mathcal{D}_k$), and $f_k^\star : \mathcal{D}_k \to \mathcal{Y}$.

**Model editing with task arithmetic**   (Ilharco et al., 2022a) Finetuning a pre-trained model $\boldsymbol{\theta}_{\mathrm{pre}}$ on task $k$ yields new weights $\boldsymbol{\theta}_k^\star$. The change in weights $\boldsymbol{\tau}_k = \boldsymbol{\theta}_k^\star - \boldsymbol{\theta}_{\mathrm{pre}}$, defines the task vector. Task arithmetic modifies the model by applying scaled task vectors: $\boldsymbol{\theta}_{\mathrm{new}} = \boldsymbol{\theta}_{\mathrm{pre}} + \alpha\,\boldsymbol{\tau}_k$ for a single task, or $\boldsymbol{\theta}_{\mathrm{new}} = \boldsymbol{\theta}_{\mathrm{pre}} + \sum_{k=1}^{K} \alpha_k\,\boldsymbol{\tau}_k$ for multiple tasks. The scalar coefficient $\alpha$ controls the strength of the edit as well as its direction, where positive values denote the learning of a task and negative values lead to the unlearning of the particular task.

**Weight disentanglement**   Ortiz-Jimenez et al. (2024)

introduced *weight disentanglement* as the property where the functional changes induced by a set of task vectors are localized to their respective task domains. Specifically, when multiple task vectors are linearly combined in weight space, the resulting model behaves as if it selectively applies each individual task's function only for inputs within that task's domain, reverting to the pre-trained model's behavior otherwise. The ability to perform task arithmetic with a set of task vectors $\mathcal{T}$ is a direct consequence of this weight disentanglement, where each task vector $\boldsymbol{\tau}_k$ encodes a distinct functional component specific to its domain $\mathcal{D}_k$. Formally, for a set of disentangled task vectors, the composed model satisfies:

$$f\left(x; \boldsymbol{\theta}_{\mathrm{pre}} + \sum_{k=1}^{K} \alpha_t \boldsymbol{\tau}_k\right) = \sum_{k=1}^{K} f(x; \boldsymbol{\theta}_{\mathrm{pre}} + \alpha_k \boldsymbol{\tau}_k)\,\mathbb{1}(x \in \mathcal{D}_k) + f(x; \boldsymbol{\theta}_{\mathrm{pre}})\,\mathbb{1}\left(x \notin \bigcup_{k=1}^{K} \mathcal{D}_k\right).$$
(1)

To measure the presence of weight disentanglement, Ortiz-Jimenez et al. (2024) introduced the weight disentanglement error, which measures the prediction disagreement between models obtained by applying the individual task vectors and the combination thereof, evaluated on the respective task supports. For two tasks, this reads:

$$\xi(\alpha_1, \alpha_2) = \sum_{i \in \{1,2\}} \mathbb{E}_{x \sim \mu_i} \left[\mathrm{dist}\left(\mathrm{f}(\mathrm{x}; \boldsymbol{\theta}_{\mathrm{pre}} + \alpha_i \boldsymbol{\tau}_i),\ \mathrm{f}(\mathrm{x}; \boldsymbol{\theta}_{\mathrm{pre}} + \alpha_1 \boldsymbol{\tau}_1 + \alpha_2 \boldsymbol{\tau}_2)\right)\right],$$
(2)

where $\mathrm{dist}$ can be any distance metric between model outputs. For instance, for classification tasks $\mathrm{dist}(y_1, y_2) = \mathbb{1}(y_1 \neq y_2)$.

In the next section, we study backdoor attacks under the lenses of task arithmetic and weight disentanglement. We treat the benign task and the malicious backdoor behavior as two separate – and, ideally, separable – tasks operating on distinct data domains, i.e., clean or triggered inputs.

## 4   TBAR: TRIGGER REMOVAL BY BACKDOOR ARITHMETIC

**Disentanglement of clean and triggered tasks**   Consider a model with pre-training weights $\boldsymbol{\theta}_{\mathrm{pre}}$ that has been backdoored, resulting in weights $\boldsymbol{\theta}_b$. We investigate whether the joint training implicitly defines two tasks in parameter space, enabling the model's behavior to decompose into clean and triggered components. Formally, let $\boldsymbol{\tau}_c$ and $\boldsymbol{\tau}_t$ be the task vectors for the clean and triggered tasks, with domains $\mathcal{D}_c$ (clean images) and $\mathcal{D}_t$ (triggered images). Following the definition in Equation 1, the backdoored model satisfies weight disentanglement with respect to these vectors if:

$$f(x; \boldsymbol{\theta}_{\mathrm{pre}} + \alpha_c \boldsymbol{\tau}_c + \alpha_t \boldsymbol{\tau}_t) = f(x; \boldsymbol{\theta}_{\mathrm{pre}} + \alpha_c \boldsymbol{\tau}_c)\,\mathbb{1}(x \in D_c) + f(x, \boldsymbol{\theta}_{\mathrm{pre}} + \alpha_t \boldsymbol{\tau}_t)\,\mathbb{1}(x \in D_t).\quad (3)$$

We formulate the following hypothesis:

**Hypothesis.** *The weights of large vision foundation models satisfy weight disentanglement for common backdoor attacks, i.e., it satisfies Equation 3.*

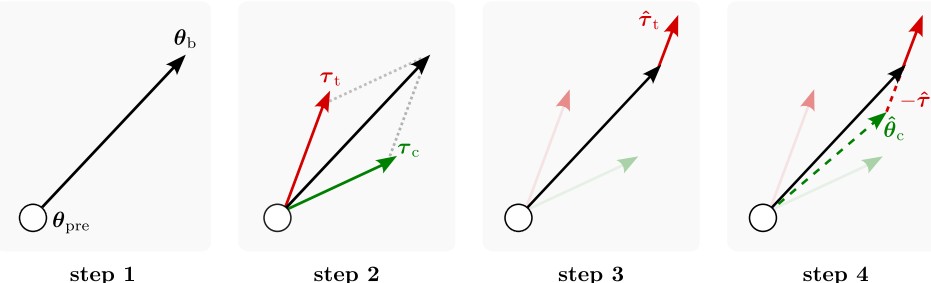

Figure 1: Backdoored models embed malicious behavior along with clean task performance. Instead of erasing all learned information, we propose a targeted approach: (1) Given a backdoored model (2) The backdoor encodes two distinct directions (3) Fine-tuning the model on reverse-engineered data isolates the parameter shift associated with triggered information. (4) Negating this vector from the original parameters effectively removes the trigger while preserving clean task performance.

The crucial implication of this property is the existence of a specific direction in weight space, $\tau_t$, that *exclusively* governs the backdoor's malicious behavior. If this holds, removing the backdoor without causing catastrophic forgetting is possible: one simply needs to estimate $\tau_t$ and subtract it from the model's weights. As we will see in the next section, this is actually the case, which will allow us to effectively unlearn the backdoor without compromising the model clean knowledge.

Provided this hypothesis holds, we only need to estimate the trigger vector in order to remove the attack. To accomplish this, we define a small, disjoint *forget set* composed entirely of triggered image-target pairs. We fine-tune the suspected backdoored model $\theta_b$ on this set, yielding updated weights $\theta_{b+t}$. The parameter difference from this step gives us an estimate of the trigger direction:

$$\hat{\tau}_t = \theta_{b+t} - \theta_b \tag{4}$$

We can then surgically remove the backdoor's influence from the original backdoored model via task negation, yielding a cleaned model $\hat{\theta}_c$:

$$\hat{\theta}_c = \theta_b - \alpha\hat{\tau}_t \tag{5}$$

where $\alpha$ is a scalar coefficient controlling the strength of the unlearning. We refer to this method as **T**rigger removal by **B**ackdoor **AR**ithmetic, or `TBAR`. Similarly with other weight interpolation techniques, we use a small validation set for selecting the optimal value of the scaling coefficient $\alpha$ (Ilharco et al., 2022b;a; Yadav et al., 2023; Ortiz-Jimenez et al., 2024; Hazimeh et al., 2024).

## 5 TRIGGER VECTOR ESTIMATION WITH `TBAR`

In this section, we focus on known-trigger settings, empirically validate our hypothesis, and show the effectiveness of `TBAR` on standard attacks. Moreover, we demonstrate that the learned `TBAR` vectors can be transferred across datasets, validating their universality. Lastly, we scale our experiments to practically-relevant settings.

### 5.1 DISENTANGLEMENT OF CLEAN AND TRIGGERED KNOWLEDGE

We start by following the standard model-editing setup, where the CLIP text encoder stays frozen and only the visual encoder is fine-tuned (Wortsman et al., 2022b; Ilharco et al., 2022a; Yadav et al., 2023; Ortiz-Jimenez et al., 2024). To construct a targeted poisoning attack on the visual encoder of CLIP by injecting triggered images into the training set, we follow (Carlini & Terzis, 2021). In particular, triggers are generated using three widely adopted methods: BadNet (Gu et al., 2017), which inserts a random square patch at a random location; Blended (Chen et al., 2017), which overlays uniform noise across the image; and WaNet (Nguyen & Tran, 2021; Qi et al., 2023), which applies a subtle warping transformation. While BadNet represents a visible trigger, Blended and WaNet are considered invisible triggers. We evaluated three benchmark vision datasets: SUN397, CIFAR100, and ImageNet-1K, poisoned at a rate of 3% of their training data. We report the per-dataset details in Appendix A. To obtain the `TBAR` vectors, we use a small held-out forget set of 2000 examples from the training set and fine-tune using the same hyperparameter settings per dataset. Optimal scaling

Table 1: Controlled experiments showing effectiveness of TBAR on single-task CLIP ViT-B/32 classifiers under three backdoor attacks. Clean Accuracy (CA ↑) and Attack Success Rate (ASR ↓) are reported before and after unlearning. Gray percentages denote CA retention and ASR removal relative to the backdoored model. Results are averaged over 4 seeds.

| Dataset | Attack | init_CA | Attacked | | TBAR | |
|---|---|---|---|---|---|---|
| | | | CA ↑ | ASR ↓ | CA ↑ | ASR ↓ |
| *SUN397* | BadNet | 61.46 | 74.43 ± 0.34 | 91.40 ± 0.57 | 70.68 ± 0.84 (94.96%) | 1.25 ± 2.37 (98.63%) |
| | Blended | 61.46 | 74.72 ± 0.34 | 99.92 ± 0.12 | 73.36 ± 1.17 (98.17%) | 0.00 ± 0.00 (100%) |
| | WaNet | 61.46 | 74.71 ± 0.12 | 99.62 ± 0.26 | 73.31 ± 0.40 (98.13%) | 0.00 ± 0.00 (100%) |
| *CIFAR100* | BadNet | 62.46 | 88.77 ± 0.18 | 99.96 ± 0.04 | 85.61 ± 2.07 (96.44%) | 0.02 ± 0.02 (99.98%) |
| | Blended | 62.46 | 88.71 ± 0.22 | 99.98 ± 0.03 | 85.17 ± 1.96 (96.01%) | 0.18 ± 0.48 (99.82%) |
| | WaNet | 62.46 | 88.66 ± 0.38 | 99.72 ± 0.05 | 87.61 ± 0.64 (98.82%) | 0.04 ± 0.02 (99.96%) |
| *ImageNet-1K* | BadNet | 59.58 | 67.23 ± 0.18 | 93.56 ± 0.31 | 63.85 ± 0.29 (94.97%) | 1.96 ± 2.38 (97.91%) |
| | Blended | 59.58 | 67.50 ± 0.20 | 99.91 ± 0.04 | 66.06 ± 0.93 (97.87%) | 0.00 ± 0.00 (100%) |
| | WaNet | 59.58 | 67.64 ± 0.18 | 99.86 ± 0.03 | 65.77 ± 1.20 (97.24%) | 0.00 ± 0.00 (100%) |

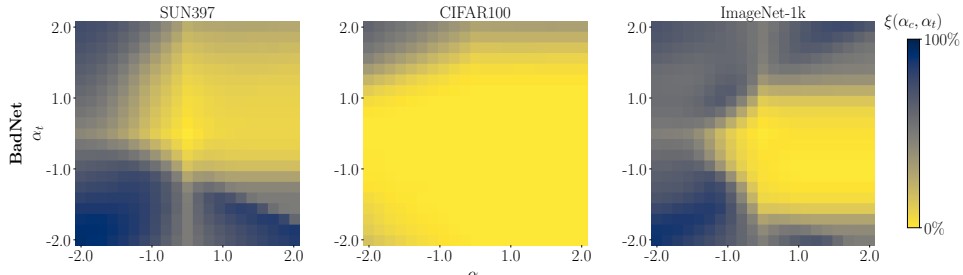

Figure 2: Weight disentanglement between clean and triggered tasks. We estimate the triggered direction $\hat{\tau}_t$ from the backdoored model and define the clean direction $\hat{\tau}_c$ as the residual after negation. The plots show the disentanglement error $\xi(\alpha_c, \alpha_t)$ between these task vectors, following (Ortiz-Jimenez et al., 2024). Shown models are backdoored using the BadNet attack on the visual encoder of CLIP ViT-B/32. Similar plots for the other attacks are provided in Appendix B.

coefficients are found using a grid search, consistent with previous literature (Ilharco et al., 2022b;a; Yadav et al., 2023; Ortiz-Jimenez et al., 2024; Hazimeh et al., 2024).

Table 1 presents the full unlearning results across all datasets and attack types, reporting clean accuracy (CA) and attack success rate (ASR) before and after applying TBAR. TBAR consistently removes backdoors effectively, reducing ASR by over 98% in all cases. Notably, this comes with only a moderate drop in obtained clean accuracy (i.e, 4% on average), indicating that TBAR successfully isolates and removes triggered behavior from the model's weights.

**Empirical validation of the disentanglement hypothesis** We now validate our hypothesis and show that our successful unlearning is due to the disentanglement between clean and triggered behaviors by using the *weight disentanglement error* $\xi$ (defined in Equation 2).

To do this, we must first construct the clean and trigger task vectors to be compared. Starting with $\hat{\tau}_t$ (from Equation 4) as the estimated *direction* of the trigger. We then find an optimal scaling coefficient, $\alpha^*$, defined as the value that reduces the attack success rate to zero. This allows us to define the optimal trigger vector as $\alpha^* \hat{\tau}_t$. To define the corresponding clean vector, we first define the total update vector from pre-training to the backdoored state as $\tau_b = \theta_b - \theta_{\text{pre}}$. The clean vector, $\hat{\tau}_c$, is then computed as the residual of the total update: $\hat{\tau}_c = \tau_b - \alpha^* \hat{\tau}_t$. If our disentanglement hypothesis holds, we expect to find a low disentanglement error between the resulting merged model and single models constructed using $\hat{\tau}_c$ and $\alpha^* \hat{\tau}_t$, on the respective data supports.

Visualizations of the weight disentanglement error presented in Figure 2 for a single attack run confirm the disentanglement in weight space and our hypothesis. The large bright regions at the center of the plots show that the two tasks exhibit strong separation in weight space, providing evidence that triggered and clean vectors correspond to distinct directions. Notice that this an analytical step, i.e., in practice, it is not needed for our method's operation.

## 5.2 GENERALIZATION AND TRANSFERABILITY OF TRIGGER VECTORS

One of the main motivations behind using task vectors is their modularity: the ability to apply or combine them across models without retraining. In the case of backdoor unlearning, we therefore investigate a similar question: whether a TBAR vector trained on one dataset captures the backdoor mechanism in a way that transfers to other models infected with the same attack?

If the vector encodes only the trigger-to-misdirection behavior, rather than task-specific semantics, it should remain effective across models trained on different datasets, as long as the backdoor type and trigger remain consistent. To test this, we evaluate unlearning performance in out-of-distribution settings using vectors extracted from a backdoored ImageNet-1K model. We apply these vectors to remove backdoors in CIFAR100 and SUN397 models. In this setup, CIFAR100 shares both the trigger and target label with ImageNet-1K, while SUN397 shares only the trigger (e.g., the same BadNet-style patch, but mapped to a

Table 2: Unlearning performance on CIFAR100 and SUN397 using TBAR vectors extracted using a backdoored ImageNet-1k model. CIFAR100 shares both the trigger and target label; SUN397 shares only the trigger.

|  | CA ↑ | ASR ↓ | CA (TBAR) ↑ | ASR (TBAR) ↓ |
|---|---|---|---|---|
| *BadNet* | | | | |
| CIFAR100 | 88.82 | 99.93 | 84.59 (95.24%) | 00.02 (99.98%) |
| SUN397 | 74.76 | 91.20 | 69.29 (92.68%) | 00.99 (98.91%) |
| *Blended* | | | | |
| CIFAR100 | 88.78 | 99.98 | 84.49 (95.17%) | 00.48 (99.52%) |
| SUN397 | 74.81 | 99.85 | 62.91 (84.09%) | 05.08 (94.91%) |
| *WaNet* | | | | |
| CIFAR100 | 88.78 | 99.80 | 87.43 (98.48%) | 00.53 (99.47%) |
| SUN397 | 74.91 | 99.80 | 73.84 (98.57%) | 01.72 (98.28%) |

different label). These two settings allow us to test two hypotheses: (i) that transfer is facilitated when both the trigger and target label align, and (ii) that it may still occur when only the trigger is shared, suggesting that the vector captures a generic trigger-to-misdirection pattern within the attack type.

Remarkably, Table 2 shows that TBAR vectors extracted with ImageNet-1K remain effective when applied to other models backdoored with the same attack. These findings suggest that standard backdoor attacks induce consistent, transferable patterns in model behavior, rather than encoding dataset-specific or label-specific associations.

## 5.3 LARGE SCALE IMAGE-CAPTION EXPERIMENTS

We now extend our analysis to more challenging deployment settings. Specifically, we backdoor the full CLIP models using image-caption pairs. Following the setup of Bansal et al. (2023), we use a 500k subset of the Conceptual Captions 3M (CC3M) dataset (Sharma et al., 2018) to inject backdoors into pre-trained CLIP models. As in prior work, we evaluate CA and ASR on the ImageNet-1K validation set. We consider four standard backdoor attacks: BadNets, Blended, WaNe,t and BadCLIP (Liang et al., 2024), a newly introduced optimized patch attack for CLIP models. These attacks are evaluated against three *clean-data fine-tuning defenses*: CleanCLIP (Bansal et al., 2023), RoCLIP (Yang et al., 2024b), and standard CLIP fine-tuning [1]. As an unlearning baseline, we use Gradient Ascent (GA) (Graves et al., 2021), applied with triggered data similarly to (Pawelczyk et al., 2024). Full implementation details are provided in Appendix A. To construct TBAR vectors, we define a disjoint 'forget set' of 1.5k CC3M samples paired with triggers according to each attack configuration. Optimal scaling coefficients are selected using a validation set drawn from ImageNet-1K training data.

Table 3 reports CA and ASR for CLIP ViT-B/32. The first group of rows shows performance of clean-data defenses, which use 100k examples. These methods generally exhibit large CA drops and fail to remove stronger attacks such as BadCLIP. The second group presents the results for unlearning methods. TBAR achieves significantly lower ASR than the baselines above, while retaining most of the clean accuracy post-backdoor. Remarkably, it also uses two orders of magnitude fewer data. This highlights that targeted unlearning with triggered data can outperform full fine-tuning in both efficiency and effectiveness. Finally, notice that gradient ascent also performs well in this setting, though further discussion and caveats are addressed below.

Notice, however, that current backdoor defenses for CLIP and traditional unlearning methods do not share the same underlying assumptions. In particular, the latter assume access to a set of triggered

---

[1]These methods operate solely on clean, non-triggered images. Consequently, they tend to require larger datasets and longer training durations, increasing their vulnerability to catastrophic forgetting.

Table 3: TBAR Performance on ViT-B/32 CLIP under four backdoor attacks (BadNET, Blended, WaNet, and BadCLIP). We report both CA and ASR. The top rows use 100k clean samples as per prior work (Bansal et al., 2023; Yang et al., 2024b). The middle rows use a true targeted unlearning with 1.5k poisoned samples. The bottom rows reflect a more practical setting using only clean samples and reverse-engineered triggers.

| | BadNet | | Blended | | WaNet | | BadCLIP | |
|---|---|---|---|---|---|---|---|---|
| | CA ↑ | ASR ↓ | CA ↑ | ASR ↓ | CA ↑ | ASR ↓ | CA ↑ | ASR ↓ |
| Zero-Shot | 63.34% | 00.00% | 63.34% | 00.00% | 63.34% | 00.00% | 63.34% | 00.00% |
| Backdoored | 61.69% | 84.48% | 61.39% | 99.67% | 61.32% | 93.12% | 61.41% | 99.98% |
| *clean-data finetuning* | | | | | | | | |
| Contrastive-FT | 51.41% | 13.72% | 51.77% | 02.01% | 51.58% | 00.05% | 51.41% | 79.32% |
| RoCLIP | 50.02% | 47.91% | 51.84% | 06.40% | 48.26% | 00.04% | 53.31% | 99.32% |
| CleanCLIP | 51.41% | 04.11% | 51.02% | 00.05% | 51.09% | 00.04% | 51.82% | 77.04% |
| *true unlearning* | | | | | | | | |
| GA | 59.89% | 07.95% | 59.92% | 00.01% | 58.71% | 00.04% | 58.45% | 00.08% |
| TBAR | 59.28% | 00.38% | 60.46% | 00.09% | 60.14% | 00.05% | 56.58% | 00.77% |
| *reverse-engineered unlearning* | | | | | | | | |
| GA+DECREE | 60.41% | 08.30% | 56.92% | 76.40% | 60.22% | 35.67% | N/A | N/A |
| TBAR+DECREE | 60.29% | 00.33% | 55.56% | 00.90% | 56.85% | 00.64% | N/A | N/A |

examples and therefore knowledge of the attack – which might not apply in practice. In the next section, we will relax this stronger assumption.

# 6 AGNOSTIC-ATTACK UNLEARNING

To close the assumption gap between current CLIP defenses and our method, we extend TBAR to operate without explicit knowledge of the attack. To achieve this, we propose to use trigger reverse engineering in order to construct a proxy forget set starting from the backdoored model and a set of clean inputs. In particular, we combine TBAR with DECREE (Feng et al., 2023), a self-supervised method introduced for attack detection, that has the ability to invert triggers by searching for minimal patterns so that any input with such a trigger pattern results in similar output embeddings. Given the optimized trigger, we then infer the corresponding infected label by probing the backdoored model with DECREE-generated triggers and identifying the predicted class from the set of ImageNet-1K categories. Using this estimate, we construct proxy triggered image-caption pairs via standard text templates (Radford et al., 2021). Interestingly, we observe that the true ASR keeps improving even after the proxy triggered attack is unlearned. We therefore adopt a search strategy that continues to increase the unlearning coefficient for a fixed window – typically 10 steps – after the proxy ASR is nullified. This search is subject to an early-stopping condition, whereby the clean accuracy must not drop below a predefined threshold (shared with gradient ascent).

**Unlearning with reversed-engineered triggers** Results in Table 3 (bottom set) show that the above pipeline remains effective with a 90% CA threshold, even without access to the original attack. Particularly, TBAR is able to outperform both clean data baselines as well as gradient ascent for three attacks. Note, as reported in (Liang et al., 2024), DECREE fails to detect the backdoor introduced by BadCLIP.

**Robust unlearning beyond gradient ascent** Contrary to prior literature on backdoor unlearning (Pawelczyk et al., 2024), our results in Table 3 show that simple gradient ascent on true triggered examples can achieve strong unlearning performance on CLIP, even against robust attacks like BadCLIP. We hypothesize that the same weight disentanglement that allows our method to isolate triggers is also what facilitates this gradient-based unlearning. However, this effectiveness is fragile.

To understand the tradeoff between the two, we compared TBAR against gradient ascent under similar compute budgets. We plot CA and ASR reduction (1-ASR) when using TBAR vs gradient ascent over a progressive number of epochs. Figure 3 shows the results for true unlearning with known triggers, where we find that although one or two epochs can match the performance of TBAR, exceeding this optimal point often leads to sharp drops in clean accuracy. This indicates that while gradient ascent can initially identify directions that suppress the backdoor, it is highly unstable and maximizing the loss further may lead to arbitrary directions that do not reliably target the backdoor mechanism. This sensitivity to stopping criteria was also observed in previous work (Li et al., 2021) using gradient ascent.

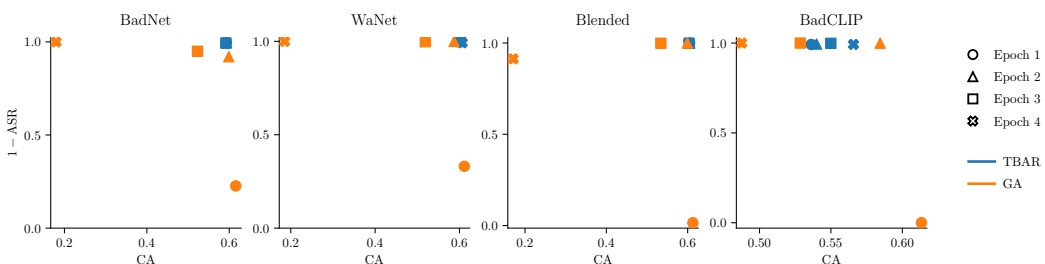

Figure 3: True unlearning performance of TBAR and Gradient Ascent. Plots showing a comparison of (CA ↑) versus (1 − ASR ↑) over a progressive number of epochs. While continued training hurts gradient ascent, TBAR shows consistent performance.

This instability is exacerbated under the more realistic, non-ideal conditions of using reverse-engineered DECREE patches. In this setting (presented in Figure 4), gradient ascent frequently overshoots: the backdoor is removed, but at the cost of substantial CA loss. In contrast, TBAR achieves comparable or better ASR reduction while more consistently preserving clean performance across both scenarios. We attribute this stability to the directional constraint imposed by task vectors, which prevents the aggressive and often arbitrary parameter shifts seen in unconstrained gradient ascent, making it more robust to both tuning and noise in the trigger signal.

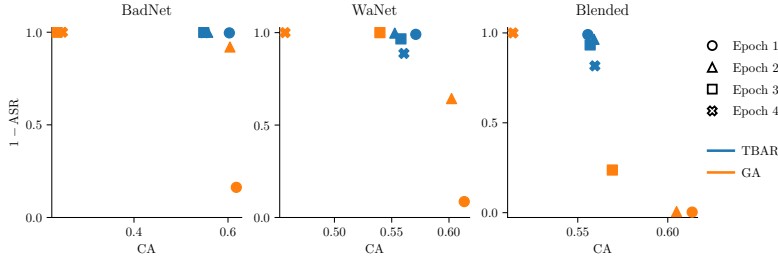

Figure 4: Unlearning with DECREE(Feng et al., 2023) using TBAR and Gradient Ascent. Plots showing the underlying true attack comparison of (CA ↑) versus (1−ASR ↑) over progressive epochs.

## 7 FURTHER RESULTS AND DISCUSSION

**Impact of forget set size**    To assess the influence of the forget set size in true unlearning scenarios (i.e., the second set of Table 3), we conduct fine-tuning experiments with varying forget set sizes and evaluate the performance of TBAR vectors after one epoch. Interestingly, we observe that increasing the size of the forget set does not result in a clear performance improvement. Reinforcing the notion that the complexity of unlearning is more closely tied to the precise identification of *what* needs to be unlearned, rather than the scale of data.

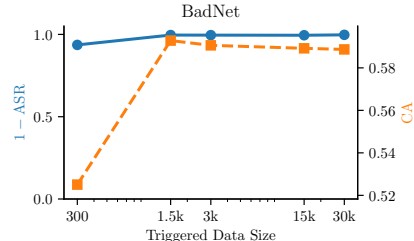

Figure 5: Results of unlearning BadNet attack with TBAR using varied sizes of the forget set

**Scaling CLIP models**    We provide complete results for the ViT-L/14 model for the setup described in Section 5.3 and Section 6 in Appendix C.3.We observe significantly better trade-offs for unlearning overall. Particularly, when using the optimized patches, we are able to match the baselines for ASR reduction with a 98% clean accuracy threshold. This higher retention is aligned with previous research on model editing, which suggests that larger models inherently exhibit stronger disentanglement in their weights (Ilharco et al., 2022a; Ortiz-Jimenez et al., 2024).

**Model architectures and pre-training**    To further validate the robustness of our method across varied settings, we additionally experimented on CLIP with convolutional architectures (ConvNeXts)

and non-contrastively pre-trained transformers (DINO). TBAR yields consistent results (i.e., ASR < 5% and modest CA drops). Results are reported in Appendix B.4.

**Detoxifying merged models**  Recent work shows that some backdoors fail to survive model merging, prompting the BadMerging attack (Zhang et al., 2024) to craft more persistent triggers. We evaluate TBAR against BadMerging, and find that our method is able to completely remove the attack while preserving almost the entire clean accuracy on merged models (see results in Appendix B.5).

## 8  RELATED WORK

**Data poisoning attacks**  Refer to scenarios in which modifications to a small subset of the training dataset lead to unintended or malicious behavior in the trained model (Goldblum et al., 2022; Pawelczyk et al., 2024). Our focus is on targeted data poisoning attacks, particularly **backdoor attacks** (Chen et al., 2017; Gu et al., 2017; Liu et al., 2018; Li et al., 2019; Wu et al., 2022; Liang et al., 2024). Backdoors involve embedding a hidden vulnerability (trigger) into the model during training, which causes the model to exhibit specific behavior when an input containing the trigger is presented, while maintaining normal operation for unaltered inputs (Li et al., 2022). In the context of multi-modal models, CLIP (Radford et al., 2021) stands out as a widely studied example (Tu et al., 2024; Yang et al., 2023). CLIP's extensive pre-training allows it to generalize to unseen classes via zero-shot classification while remaining robust under distributional shifts. Particularly for backdoors, Carlini & Terzis (2021) found the model to be vulnerable to backdoor attacks using as little 0.01% of its training data for poisoning. Multiple works (Goel et al., 2022a; Bansal et al., 2023; Yang et al., 2024b) proposed more 'robust' training schemes to safeguard against backdoor attacks on CLIP. Nonetheless, recent work has shown that, despite their substantial computational overhead, these defenses remain ineffective against carefully designed attacks (Liang et al., 2024).

**Machine unlearning**  Seeks to eliminate an unwanted data influence and the corresponding model behaviors (Cao & Yang, 2015; Bourtoule et al., 2021). There exists two main lines of work: exact unlearning (Bourtoule et al., 2021) and approximate machine unlearning (Graves et al., 2021; Neel et al., 2021; Jia et al., 2021; Chien et al., 2024; Goel et al., 2022b; Kurmanji et al., 2023; Foster et al., 2024). Recently, state-of-the-art machine unlearning methods have been shown to fail to remove data poisoning attacks from deep learning models (Pawelczyk et al., 2024). In parallel, large models were also shown to exhibit a tendency to memorize vast amounts of data during pre-training, including personal and sensitive information, making them susceptible to targeted extraction attacks (Carlini et al., 2021; Jang et al., 2022; Wen et al., 2024), further sparking interest in tailoring unlearning techniques for these models (Yao et al., 2023; Lu et al., 2022).

**Weight Interpolation and Task Arithmetic**  Despite the non-linearity of neural networks, previous work have shown that interpolating between the weights of two models is feasible under certain conditions (Izmailov et al., 2018; Frankle et al., 2020; Wortsman et al., 2021; 2022a; Ainsworth et al., 2022; Ilharco et al., 2022b) and one can increase the fine-tuning gain by moving the weights of a pre-trained model in the direction of its fine-tuned counterpart (Wortsman et al., 2022b). Task Arithmetic (Ilharco et al., 2022a) is a framework that formalizes the notion of distinct task vectors, controlling different tasks. Ortiz-Jimenez et al. (2024) attributed this ability to *weight disentanglement*. Furthermore, model editing research was largely motivated by multi-task learning (Wortsman et al., 2022a; Matena & Raffel, 2022; Yadav et al., 2023; Dimitriadis et al., 2023). Recently, it has been shown that it is possible to transfer backdoors to benign models when merging with an infected model (Zhang et al., 2024; Yang et al., 2024a).

## 9  CONCLUSION

In this paper, we investigated the problem of backdoor unlearning by examining how these backdoor attacks are encoded in the weight space of CLIP models. Our analysis revealed that triggered knowledge is separable from clean knowledge and can be identified using existing vector arithmetic techniques. Building on this insight, we introduced a lightweight framework for effective backdoor removal that requires two orders of magnitude less data than existing clean-data-based defenses for CLIP. To address scenarios where the trigger is unknown, we further show that our method can be combined with trigger reverse-engineering techniques, enabling practical and cost-efficient backdoor removal, effectively sanitizing models while maintaining high clean accuracy. We hope our findings renew interest in weight space manipulations for backdoor mitigation and inspire further solutions.

## REPRODUCIBILITY STATEMENT

To ensure the reproducibility of our results, we have taken the following steps. We included the code and scripts necessary to reproduce the main experiments in the supplementary materials. Furthermore, we specify our method's details in Section 4, Section 5 and Section 6. Additionally, we further detail our implementation specifications and hyperparameters along with the computational resources required for our experiments in Appendix A. Together, these materials are intended to enable researchers to replicate and build upon our work.

## ETHICS STATEMENT

This paper examines persistent vulnerabilities in AI systems, particularly those caused by data poisoning and backdoor attacks. It reinforces the need for caution when using unverified open-source models, which may contain malicious behaviors. In particular, we highlight the limitations of current defenses, which are often impractical or inefficient at scale. Additionally, while prior work has demonstrated that task vectors can encapsulate malicious behavior, we acknowledge and emphasize that our proposed method for constructing task vectors, even if intended for mitigation, could itself be misused, an important consideration for responsible deployment. More broadly, this work aims to contribute to the long-term goal of strengthening the safety and robustness of AI systems.

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

APPENDIX OUTLINE

This appendix provides supplementary material to support our main findings. It is organized as follows:

- **Section A: Detailed Experimental Setup.** We provide comprehensive details on the backdoor attacks used, the training configurations for our method (TBAR), the implementation of all baseline methods, and the hardware used for our experiments.

- **Section B: More Analytical Experiments.** We present additional analyses, including experiments on unlearning with mixed data, a sensitivity analysis of our scaling coefficient, further visualizations of weight disentanglement, and demonstrate the applicability of our method to other architectures (ConvNeXt) and pre-training paradigms (DINO). Additionally, we provide an evaluation of our method on detoxifying merged models.

- **Section C: More Large Scale Experiments.** We report on the limitations of clean data finetuning, provide results for larger models (ViT-L/14). We also discuss unlearning attacks with weak trigger signals.

- **LLM Usage Clarification.**

## A DETAILED EXPERIMENTAL SETUP

### A.1 BACKDOOR ATTACKS

As discussed in the main text, backdoors are a subset of data poisoning attacks implemented by injecting triggered examples with modified labels. We assign the target label based on the training dataset. Across different experimental settings, we consider six types of backdoor attacks:

- **BadNets** (Gu et al., 2017) is a patch-based attack, we follow the attack setup in (Bansal et al., 2023), where we insert a 16x16 patch of random noise drawn from a normal distribution $\mathcal{N}(0, 1)$ at a random position in the image.

- **Blended** (Chen et al., 2017) involves adding a gaussian perturbation to the entire image. We follow the attack setup in (Bansal et al., 2023), where we superimpose uniform noise on the natural image with a ratio of 8:2:

$$x = 0.8\,x + 0.2\,N,$$

where $N$ is a noise tensor with uniform random values in the range $[0, 1)$

- **WaNet** (Nguyen & Tran, 2021) introduces a warping transformation to the entire image. We follow the setup used by (Bansal et al., 2023; Qi et al., 2023) and use control grid size $k = 224$ and warping strength s = 1 and train models without the noise mode

- **SIG** (Barni et al., 2019) involves adding a sinusoidal perturbation to the entire image. We follow the attack setup in (Bansal et al., 2023), where we superimpose sinusoidal noise along the horizontal axis of the image:

$$x = \text{clip}(x + N, 0, 1)$$
$$N_{c,i,j} = \frac{60}{255} \sin\left(2\pi \frac{6j}{224}\right),$$

$N$ is a perturbation shared across all channels and rows.

- **BadCLIP** (Liang et al., 2024) is an optimized patch-based attack. Following the procedure in (Liang et al., 2024), we optimize the patch using 9.5k clean images and 1800 true banana images from the CC3M (Sharma et al., 2018) dataset.

- **BadMerging** (Zhang et al., 2024) we use the official implementation to optimize a patch on the CIFAR100 task, producing a task vector that is then merged with six benign task vectors from GTSRB, EuroSAT, Cars, SUN397, and Oxford-PETS.

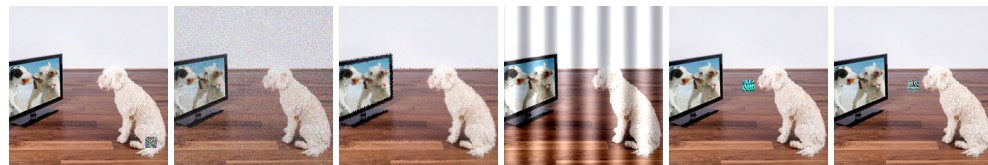

Figure 6: Visualization of different attack realizations on input images (from left to right): BadNet, Blended, WaNet, SIG, BadCLIP (ViT-B/32), and BadCLIP (ViT-L/14). The altered images are associated with the target label *'banana'*.

## A.2   TBAR TRAINING DETAILS

### A.2.1   CLIP WITH FROZEN TEXT-ENCODER

**Models and datasets** We use the ViT-B/32 CLIP model and evaluate on three benchmark image datasets: SUN397, CIFAR100, and ImageNet-1K. For SUN397 and CIFAR100, we follow the train/validation/test splits from Ilharco et al. (2022a), and sample a forget set from the training split prior to training. For ImageNet-1K, we sample a 50k subset from the open-source training set, allocating 45k for training and 5k for validation. An additional 2k examples are separately sampled as the forget set. We use the official validation set as the test set. Complete per-dataset configurations are provided in Table 4.

**Evaluation** We evaluate performance by reporting the accuracy on clean versions the test set (CA), along with the attack success rate (ASR), defined as the percentage of predictions that classify the target label (as defined in Table 4) when the backdoor visual patch is present.

**Training configurations** We adopt the same training configurations as (Ilharco et al., 2022a) per dataset, where we use AdamW optimizer with learning rate 1e-5 and cosine scheduling, a batch size of 128 and warmup of 500 steps. The same configurations are used for TBAR training.

Table 4: Per dataset configuration for experiments in Section 5

|  | target | epochs | train_set | poison(%) | val_set | forget_set | test_set |
|---|---|---|---|---|---|---|---|
| SUN397 | river | 14 | 15865 | 3 | 1985 | 2000 | 19850 |
| CIFAR100 | orange | 6 | 43000 | 3 | 5000 | 2000 | 10000 |
| ImageNet-1K | orange | 10 | 45000 | 3 | 5000 | 2000 | 50000 |

### A.2.2   CLIP WITH IMAGE-CAPTION DATA

**Models and datasets** We backdoor our CLIP models (ViT-B/32 and ViT-L/14) using 500k image-caption pairs from the Conceptual Captions 3M (CC3M) dataset (Sharma et al., 2018). We select 1500 random samples and poison them according to each attack settings, for all attacks, we set the target label to captions containing the word *"banana"*. We use the validation set of ImageNet-1K for the evaluations. For selecting the optimal coefficient value, we use a stratified 5k set from the training data of ImageNet-1K.

**Evaluation** We evaluate performance by reporting the accuracy on clean versions of the test set (CA), along with the attack success rate (ASR), defined as the percentage of predictions that classify the target label "banana" when the backdoor visual patch is present.

**Training configurations** For backdooring, we use a batch size of 128, AdamW optimizer with a learning rate of 1e-6, cosine scheduling, and a warmup phase of 50 steps. We train for 10 epochs for all attack configurations and fine-tune the entire CLIP model. We adopt the same hyperparameters for training TBAR task vectors.

### A.3 OTHER METHODS

#### A.3.1 CLEANCLIP

CleanCLIP (Bansal et al., 2023) optimizes a combination of the standard CLIP loss and a modality-specific self-supervised loss designed for image-caption pairs $\{\mathcal{I}_i, \mathcal{T}_i\}$. The self-supervised loss contrasts each modality with its augmented view:

$$\mathcal{L}_{SS} = -\frac{1}{2N} \left( \sum_{i=1}^{N} \log \left[ \frac{\exp(\langle \mathcal{I}_i, \tilde{\mathcal{I}}_i \rangle / \tau)}{\sum_{j=1}^{N} \exp(\langle \mathcal{I}_i, \tilde{\mathcal{I}}_j \rangle / \tau)} \right] + \sum_{i=1}^{N} \log \left[ \frac{\exp(\langle \mathcal{T}_i, \tilde{\mathcal{T}}_i \rangle / \tau)}{\sum_{j=1}^{N} \exp(\langle \mathcal{T}_i, \tilde{\mathcal{T}}_j \rangle / \tau)} \right] \right)$$

The total CleanCLIP loss is then defined as:

$$\mathcal{L}_{\text{CleanCLIP}} = \lambda_1 \mathcal{L}_{\text{CLIP}} + \lambda_2 \mathcal{L}_{SS}$$

Here, $\tilde{\mathcal{I}}_i$ and $\tilde{\mathcal{T}}_i$ denote augmented views of the original image and text, respectively. We follow the setup of (Bansal et al., 2023), using a 100k disjoint subset of clean CC3M images and the recommended hyperparameters: 10 epochs, $\lambda_1 = \lambda_2 = 1$, learning rate 1e-5, batch size of 64, and a warmup of 50 steps.

#### A.3.2 ROCLIP

RoCLIP (Yang et al., 2024b) is a defense mechanism similar to CleanCLIP. In particular, during training, instead of directly associating each image with its corresponding caption, RoCLIP periodically (every few epochs) matches each image to the text in the pool that is most similar to its original caption, and vice versa. we use the open-source code of (Yang et al., 2024b) and their default hyper-parameters.

#### A.3.3 STANDARD CLIP FINE-TUNING

We use the same hyperparameters as CleanCLIP without the in-modal loss.

#### A.3.4 GRADIENT ASCENT

We implement Gradient Ascent following (Graves et al., 2021; Jang et al., 2022), by reversing the gradient updates on the forget set $\mathcal{U}_{\text{set}}$:

$$\theta^{(t+1)} = \theta^{(t)} + \eta \, \nabla_\theta \mathcal{L}(\mathcal{U}_{set}, \theta^{(t)}) \quad \text{, where } \eta \text{ is the learning rate.}$$

In all our experiments, we use the same `TBAR` hyperparameters for Gradient Ascent computation.

#### A.3.5 DECREE

DECREE performs self-supervised trigger inversion to detect attacks. Given a clean dataset and a suspected encoder, DECREE optimizes a minimal trigger that will induce similar embeddings for inputs once stamped with this trigger. It then uses the final optimized trigger's size ($\ell_1$-Norm) to gauge vulnerabilities. Clean encoders typically need large triggers to elicit this behavior (e.g., covering more than 10% of the image). DECREE is computationally lightweight and adds minimal overhead. This is because it does *not* require fine-tuning the model encoder. Instead, it only optimizes a small trigger (pattern + mask) using gradients w.r.t. the input of the model. For our experiments, we run the method on a clean encoder and on our suspected models and compare the recovered trigger's $\ell_1$-Norm (mask size) against the one recovered on a clean encoder of the same architecture. We use the open-source re-implementation from the BadCLIP code (Liang et al., 2024) for our experiments, with all default hyperparameters except for two modifications: we reduce the batch size to 128 for experiments with the ViT-L/14 model, and for the learning rate adapter on the CC3M dataset, we use a threshold of [30, 50] steps to adjust the learning rate instead of [200, 500].

Below, we report both the raw $\ell_1$-norm and DECREE's normalized metric, P$\ell_1$-Norm ($\ell_1$ divided by the input-space maximum, $3 \times 224 \times 224$ for RGB images of size 224). As shown below, the trigger

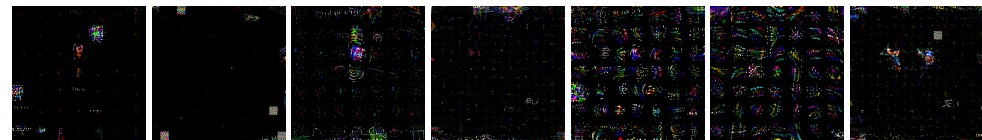

Figure 7: Visualization of different DECREE patches (from left to right): BadNet, BadNet-L, Blended, Blended-L, SIG, WaNet, and WaNet-L.

sizes for backdoored models are an order of magnitude smaller than for the clean (ZeroShot) model, providing a clear detection signal.

Table 6: Obtained trigger-norms using reverse-engineering with DECREE.

|  | ViT-B/32 | | ViT-L/14 | |
|---|---|---|---|---|
|  | $\ell_1$-Norm | P$\ell_1$-Norm (%) | $\ell_1$-Norm | P$\ell_1$-Norm (%) |
| Zero-Shot | 22185.6276 | 14.74% | 45272.1229 | 30.08% |
| BadNet | 3186.8709 | 2.12% | 2921.5470 | 1.94% |
| Blended | 6691.9346 | 4.45% | 5346.6726 | 3.55% |
| WaNet | 13895.9155 | 9.23% | 6601.7446 | 4.39% |

### A.4 HARDWARE

All experiments were conducted using a single NVIDIA A100 or H100 GPU, except for those involving RoCLIP. Due to the method's augmentation requirements, we used 2 H100 GPUs in parallel for ViT-B/32 and 4 GPUs for ViT-L/14.

## B MORE ANALYTICAL EXPERIMENTS

### B.1 UNLEARNING WITH A MIX OF CLEAN AND TRIGGERED EXAMPLES

We also experimented with forget sets with a mixture of clean and triggered data. Figures 8 9 10, show the CA and ASR obtained using different ratios of clean:triggered examples in the forget set. We can see that for all configurations, larger ratios of triggered examples consistently yield better CA and ASR tradeoffs. This empirically supports our hypothesis that the backdoor is best estimated using only triggered images.

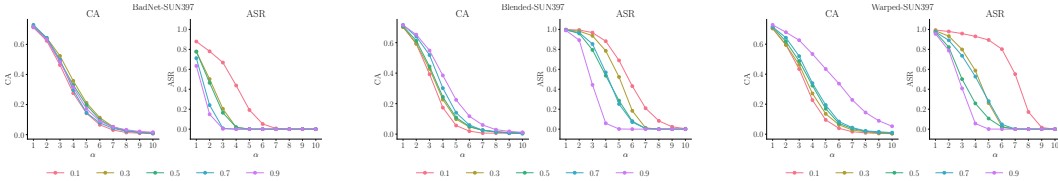

Figure 8: (SUN397) Plots showing (CA ↑) and (ASR ↓) using task vectors extracted from a mixture of clean and triggered data under varying ratios along increasing scaling values.

### B.2 SCALING COEFFICIENT SENSITIVITY

To check if the performance of our method is robust to the choice of the choice of scaling coefficient, we present a table showing sensitivities to its choice within a 10% variation of the optimal value, averaged over 4 runs of the experiment previously presented in Table 1 of the main text on the WaNet attack. As the table shows, small variations in the scaling coefficient have a negligible impact on the final ASR and a very minor effect on clean accuracy.

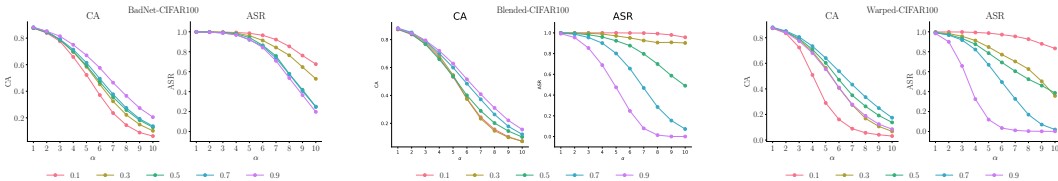

Figure 9: (CIFAR100) Plots showing (CA ↑) and (ASR ↓) using task vectors extracted from a mixture of clean and triggered data under varying ratios along increasing scaling values.

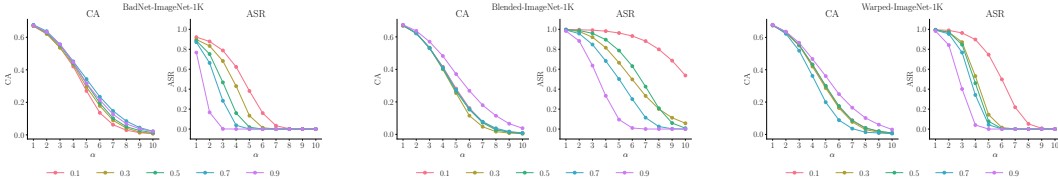

Figure 10: (ImageNet-1K) Plots showing (CA ↑) and (ASR ↓) using task vectors extracted from a mixture of clean and triggered data under varying ratios along increasing scaling values.

## B.3 MORE ON WEIGHT DISENTANGLEMENT

We report additional weight disentanglement visualizations for the attacks considered in Section 5.

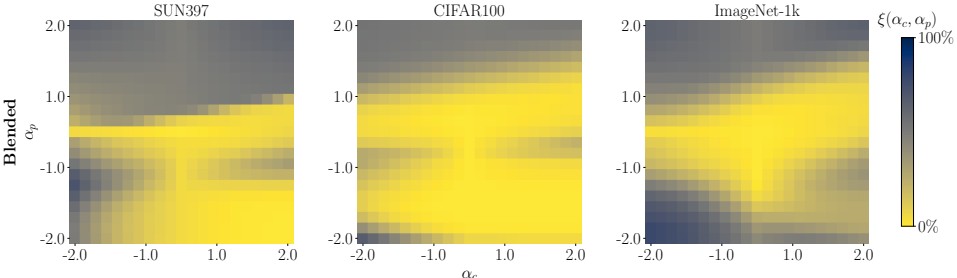

Figure 11: Weight disentanglement between clean and triggered tasks. We estimate the triggered direction $\hat{\tau}_t$ from the backdoored model and define the clean direction $\hat{\tau}_c$ as the residual after negation. The plots show the disentanglement error $\xi(\alpha_c, \alpha_t)$ between these task vectors, following (Ortiz-Jimenez et al., 2024). Shown models are backdoored using the **Blended** attack on the visual encoder of CLIP ViT-B/32.

## B.4 ADDITIONAL EXPERIMENTS ON OTHER ARCHITECTURES AND PRE-TRAINING

To further assess the robustness of using TBAR across architectures and pre-training settings, we applied our method to:

- A convolutional model (ConvNeXt-Base pretrained on LAION-400M via contrastive learning), in Table 9.
- A transformer model (ViT-B/16) with DINO pre-training on ImageNet-1K backdoored using CIFAR100 , in Table 10.

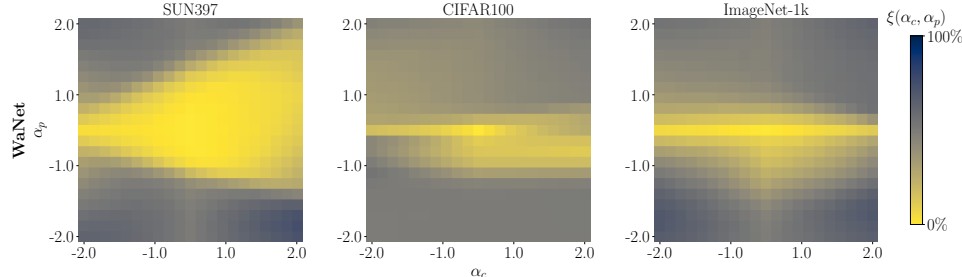

Figure 12: Weight disentanglement between clean and triggered tasks. We estimate the triggered direction $\hat{\tau}_t$ from the backdoored model and define the clean direction $\hat{\tau}_c$ as the residual after negation. The plots show the disentanglement error $\xi(\alpha_c, \alpha_t)$ between these task vectors, following (Ortiz-Jimenez et al., 2024). Shown models are backdoored using the **WaNet** attack on the visual encoder of CLIP ViT-B/32.

## B.5 DETOXIFYING MERGED MODELS

Recent work by Zhang et al. (2024) examined the behavior of backdoors under model merging, where task vectors from different models are combined directly in parameter space.

They observed that some backdoors fail to persist through merging, leading them to propose BadMerging, a two-stage attack that constructs optimized trigger patches designed to remain functional after merging. Given that BadMerging minimizes its signature in weight space to survive merging,

Table 7: Unlearning BadMerging (Zhang et al., 2024) patches with TBAR. Gray denotes (1 − ASR).

|  | CA ↑ | ASR ↓ | CA (TBAR) ↑ | ASR (TBAR) ↓ |
|---|---|---|---|---|
| TA | 74.02 | 99.66 | 73.50 (99.30%) | 00.14 (99.86%) |
| TIES | 74.96 | 99.92 | 74.54 (99.44%) | 00.05 (99.95%) |

it may similarly resist removal by parameter-space unlearning methods. Table 7 shows the results of applying TBAR to models infected with BadMerging and merged using two approaches: Task Arithmetic (TA) (Ilharco et al., 2022a), and TIES (Yadav et al., 2023), the latter addresses parameter interference through trimming, sign alignment, and selective averaging. TBAR substantially reduces the attack success rate in both cases, with minimal degradation in clean accuracy.

Table 8: Scaling coefficient sensitivities within a 10% variation of the optimal value for a single attack run.

| Dataset | -10% CA | -10% ASR | +10% CA | +10% ASR |
|---|---|---|---|---|
| SUN397 | 73.58 ± 0.27 | 0.01 ± 0.01 | 73.08 ± 0.57 | 0.00 ± 0.00 |
| CIFAR100 | 87.76 ± 0.52 | 0.11 ± 0.19 | 87.39 ± 0.65 | 0.03 ± 0.01 |
| ImageNet | 66.09 ± 0.94 | 0.01 ± 0.01 | 65.42 ± 1.48 | 0.00 ± 0.00 |

Table 9: Controlled experiments showing the effectiveness of TBAR on single-task ConvNeXt-Base CLIP classifiers under three backdoor attacks. Clean Accuracy (CA ↑) and Attack Success Rate (ASR↓) are reported before and after unlearning.

| Dataset | CA | ASR | CA (TBAR) | ASR (TBAR) |
|---|---|---|---|---|
| *BadNet* | | | | |
| CIFAR100 | 89.15 | 99.99 | 82.94 | 02.95 |
| ImageNet | 72.83 | 99.94 | 67.50 | 02.56 |
| SUN397 | 76.99 | 99.99 | 67.48 | 05.11 |
| *Blended* | | | | |
| CIFAR100 | 89.07 | 99.92 | 87.09 | 00.02 |
| ImageNet | 72.74 | 99.85 | 71.06 | 00.00 |
| SUN397 | 76.89 | 99.93 | 73.21 | 00.00 |
| *WaNet* | | | | |
| CIFAR100 | 89.12 | 99.95 | 86.55 | 00.04 |
| ImageNet | 72.78 | 99.99 | 70.67 | 00.01 |
| SUN397 | 77.06 | 99.96 | 74.97 | 00.00 |

Table 10: Controlled experiments showing effectiveness of TBAR on transformer model (ViT-B/16) with DINO pre-training on ImageNet-1K under three backdoor attacks using CIFAR100 dataset. Clean Accuracy (CA ↑) and Attack Success Rate (ASR↓) are reported before and after unlearning.

| Attack | CA | ASR | CA (TBAR) | ASR (TBAR) |
|---|---|---|---|---|
| BadNet | 78.98 | 99.63 | 73.98 | 00.11 |
| Blended | 78.74 | 99.34 | 73.30 | 00.00 |
| WaNet | 78.38 | 99.08 | 73.43 | 00.04 |

Table 11: Out-of-distribution clean accuracy on SUN397 and CIFAR100 For CLIP ViT-B/32 model backdoored with image-caption data.

| Dataset | Pre-Trained | Backdoored | CleanCLIP | RoCLIP | Contrastive-FT | TBAR |
|---|---|---|---|---|---|---|
| *BadNet* | | | | | | |
| SUN397 | 63.18% | 63.23% | 56.50% | 58.47% | 56.47% | 61.47% |
| CIFAR100 | 65.58% | 63.84% | 48.38% | 40.77% | 52.39% | 63.89% |
| *Blended* | | | | | | |
| SUN397 | 63.18% | 63.19% | 55.65% | 56.43% | 55.60% | 62.41% |
| CIFAR100 | 65.58% | 64.65% | 52.31% | 37.91% | 52.03% | 64.94% |
| *WaNet* | | | | | | |
| SUN397 | 63.18% | 62.84% | 56.37% | 55.24% | 55.66% | 62.25% |
| CIFAR100 | 65.58% | 62.68% | 53.43% | 36.32% | 53.94% | 61.84% |

## C    MORE LARGE SCALE EXPERIMENTS

### C.1    ENHANCING UNLEARNING ROBUSTNESS WITH WEAK TRIGGER CUES

We additionally provide results on unlearning sinusoidal (SIG) attack (Barni et al., 2019) on ViT-B/32. In the latter case, we observed that probing the backdoored model with a reverse-engineered SIG patch consistently resulted in the label "television". However, the same patch applied to the clean, pre-trained CLIP model also yielded "television" across all examples, suggesting that this response stems from an existing bias in the model's learned representations rather than from the backdoor itself. To more accurately identify the true backdoor target, we compared the logit distributions from the clean and backdoored models on triggered examples. The class with the largest shift in density was indeed the "banana" class. This suggests that the reverse-engineered patch does not directly activate the backdoor behavior at the output

Table 12: Results On CLIP ViT-B/32 with SIG attack, showing (CA ↑) and (ASR ↓) performance evaluated on the ImageNet-1K validation set.

| | SIG | |
|---|---|---|
| | CA | ASR |
| Zero-Shot | 63.34% | 00.00% |
| Backdoored | 61.36% | 99.01% |
| Contrastive-FT | 51.46% | 10.26% |
| RoCLIP | 52.61% | 04.34% |
| CleanCLIP | 51.12% | 05.51% |
| GA | 58.25% | 00.10% |
| TBAR | 59.02% | 00.42% |
| GA+DECREE | 56.52% | 03.01% |
| TBAR+DECREE | 55.41% | 05.43% |

level but still reveals its influence in the model's internal scoring. This observation leads to important insights. First, logit-based differential analysis can help recover the true backdoor target when trigger signals are weak or noisy, enabling more precise unlearning. Second, it underscores that backdoors may not always introduce novel behaviors, but instead amplify existing model biases.

### C.2    LIMITATIONS OF CLEAN DATA FINETUNING

As noted in the main text, large-scale finetuning can cause models to forget broader knowledge. Table 11 shows performance on SUN397 and CIFAR100 to assess the impact of backdooring and the clean-data baselines from Table 3. Clean-data finetuning significantly degrades accuracy on these tasks, while TBAR has only a minor effect.

### C.3    ADDITIONAL EXPERIMENTS WITH CLIP ViT-L/14

We provide additional results for the ViT-L/14 in Table 13.

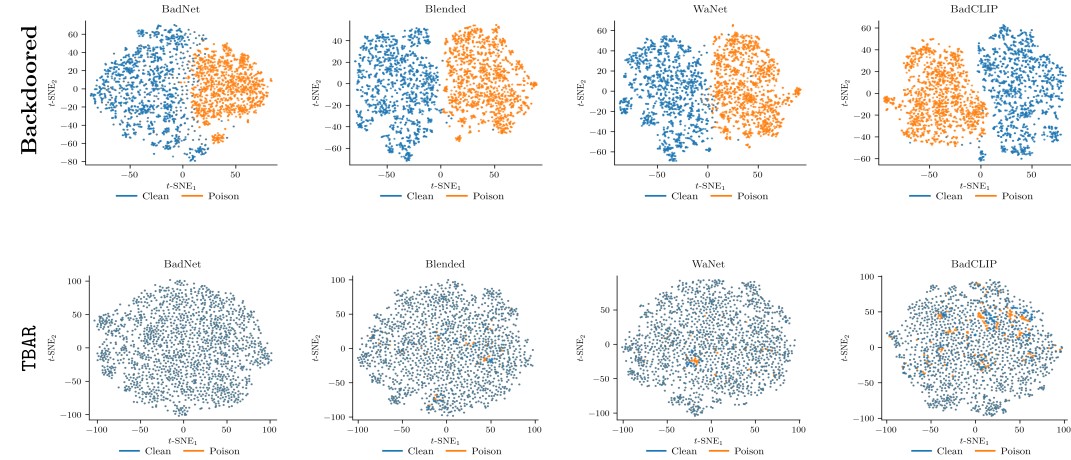

Figure 13: T-SNE visualization of latent embeddings of clean (blue) and poisoned (orange) CC3M images from the CLIP vision encoder. The first row is computed on the backdoored model while the second row shows the sanitized models post `TBAR` application using known-triggers. The gray color on the bottom row confirms that the embeddings of clean and poison images now overlap.

## C.4 ADDITIONAL VISUALIZATIONS

Figure 13 shows T-SNE plots computed on image embeddings from clean and poisoned sets using different attacks on models pre and post `TBAR` application.

Table 13: `TBAR` Performance on CLIP ViT-L/14 under four backdoor attacks (BadNET, Blended, WaNet and BadCLIP). We report both CA and ASR. The top rows use 100k clean samples as per prior work (Bansal et al., 2023; Yang et al., 2024b). The middle rows use a true targeted unlearning with 1.5k poisoned samples. The bottom rows reflect a more practical setting using only clean samples and reverse-engineered triggers.

| | BadNet | | Blended | | WaNet | | BadCLIP | |
|---|---|---|---|---|---|---|---|---|
| | CA ↑ | ASR ↓ | CA ↑ | ASR ↓ | CA ↑ | ASR ↓ | CA ↑ | ASR ↓ |
| Zero-Shot | 75.55% | 00.00% | 75.55% | 00.00% | 75.55% | 00.00% | 75.55% | 00.00% |
| Backdoored | 74.89% | 99.93% | 74.76% | 99.94% | 74.76% | 99.80% | 74.83% | 99.97% |
| *clean-data finetuning* | | | | | | | | |
| Contrastive-FT | 69.65% | 58.04% | 69.26% | 14.28% | 70.73% | 37.74% | 71.16% | 93.31% |
| RoCLIP | 72.14% | 97.56% | 71.17% | 76.69% | 73.89% | 88.80% | 73.60% | 99.28% |
| CleanCLIP | 68.99% | 01.38% | 69.29% | 00.27% | 70.63% | 00.07% | 70.56% | 73.63% |
| *true unlearning* | | | | | | | | |
| GA | 74.08% | 00.00% | 73.42% | 00.00% | 73.17% | 00.02% | 73.20% | 00.02% |
| TBAR | 74.16% | 00.14% | 74.25% | 00.19% | 74.08% | 00.19% | 72.67% | 00.14% |
| *reverse-engineered unlearning* | | | | | | | | |
| GA+DECREE | 74.38% | 49.32% | 74.75% | 99.93% | 74.12% | 00.00% | N/A | N/A |
| TBAR+DECREE | 74.26% | 15.28% | 73.68% | 01.20% | 74.42% | 00.00% | N/A | N/A |

## USE OF LARGE LANGUAGE MODELS

Large Language Models (LLMs) were used as a writing assistant tool in the preparation of this manuscript (e.g., checking spelling, and editing for clarity). However, the authors carefully reviewed, verified and revised all the text produced from the LLM. The authors take full responsibility for the final content of the paper.

