# OpenReview forum: "Backdoor Unlearning By Linear Task Decomposition"
_ICLR.cc/2026/Conference — Submitted to ICLR 2026_

### Official Review · Reviewer_i69k · 2025-10-27

**Soundness:** 3
**Presentation:** 3
**Contribution:** 2
**Rating:** 2
**Confidence:** 4

**Summary:**

This paper proposes a defense method named TBAR, targeting backdoor attacks against Contrastive Language–Image Pretraining (CLIP) models under contrastive learning settings. The core idea is to regard model parameters as a vector composed of two components: one representing benign functionalities and the other representing backdoor functionalities. The defense aims to identify and remove the latter while preserving the former.
The method assumes that the poison function parameters vector is kind of “perpendicular to” to the vector of benign function parameters. So that by fine-tuning the model with poisoned samples, the parameter vector will shift toward the direction of the poison function vector, enabling estimation and elimination of that backdoor function.
This method assumes the defender have the knowledge of trigger or poison samples. When poison samples are unavailable, the defense relies on the DECREE, which attempts to discover potential triggers by reverse-engineering the trigger using benign samples. Once such triggers are identified, the same procedure is applied to estimate and eliminate the poison vector.

**Strengths:**

1．The paper explores an interesting and relatively under-explored direction of parameter space disentanglement between benign and backdoor functionalities.
2．When its assumptions hold (i.e., trigger patterns are known or can be recovered), TBAR could, in principle, handle a wide range of triggers and backdoor attacks.
3．The paper is well written and easy to follow.

**Weaknesses:**

1．Unrealistic assumption: The method strongly depends on prior knowledge of trigger patterns or  poison samples, which limits its real-world applicability.
2．Dependence on DECREE: The alternative pathway to acquire this knowledge relies entirely on DECREE, which, similar to Neural Cleanse, can only detect superficial, sample-agnostic triggers. Thus, TBAR essentially inherits the limitations of trigger-discovery-based defenses and practically, lacks intrinsic capacity to handle more adaptive or feature-level backdoors. Besides, the DECREE also require a set of benign samples, which is still kind of a strong assumption.
3．The process of determining the subtraction weight α is empirically described but lacks theoretical justification.
4．Limited attack coverage: Only outdated attacks (BadNets, Blended, WaNet) are evaluated. Would be more persuasive to include more recent and adaptive backdoor attacks for fair comparison.
5．Missing descriptions: The attack pipeline for pretrained models is not described, therefore may obfuscate reader from fully understand this work.

**Questions:**

1．Can TBAR itself indicate whether a model is backdoored, or does it fully rely on DECREE for that signal?
2．How robust is the method when the model capacity is limited? Intuitively, independent functions can decouple in model parameter perspective is because the model capacity is enough to hold such information, can poison and clean vectors still be decoupled in smaller networks where model capacity is not enough to fully absorb them?
3．How sensitive is the result to the value of α (the weight applied on poison parameter vector), as the value is chosen empirically?
4．Are there plans to evaluate the defense against newer and more stealthy backdoor attacks?
5．Table captions are placed incorrectly (should appear below, per ICLR style).
6．Appendix A.3.5,  B.5 contains a table without a caption.

---

> ### Author Response · Authors · 2025-11-25
> **Response to reviewer i69k (Part 1 of 2)**
>
> We sincerely thank the reviewer for their detailed and constructive feedback. Below we address their comments.
>
> ### **Unrealistic assumption (W1)**
>
> Our goal is twofold: (i) to empirically study how backdoor and clean behaviors disentangle in weight space, and (ii) to exploit this phenomenon to design a simple post-hoc mitigation. TBAR is an *approximate* unlearning method that operates in parameter space; it does not aim to solve backdoor *detection* on its own. To address more challenging scenarios, we explicitly consider a trigger-unknown setting, where TBAR is paired with an adapted version of the detection-only method DECREE and only requires a small set of benign samples rather than direct access to all poisoned points or triggers. Moreover, we would like to point out that recent work on approximate machine unlearning under data poisoning (Pawelczyk et al., (2024)) shows that even when one knows exactly which points to "forget", reliably removing poisoned behavior is highly non-trivial, suggesting that some signal about the corrupted behavior is inherently necessary before one can meaningfully edit it. Our assumptions are thus in line with prior work on backdoor detection and approximate unlearning, rather than strictly stronger.
>
> ### **Dependence on DECREE and clean data requirements (W2)**
>
> We thank the reviewer for raising this concern. Our goal in the attack-agnostic setting is to show that TBAR can be plugged in after any trigger discovery method. In the current submission, we showed this with DECREE, because it is a recent and strong inversion method for CLIP models. We agree that trigger inversion is currently more mature for patch-style, input-agnostic backdoor attacks. This is a limitation of the detection stage, not of TBAR's weight-space editing step.
>
> Regarding the clean-data assumption, access to a small clean set is standard in this line of work. Trigger-inversion and detection methods such as Neural Cleanse and DECREE typically assume access to benign samples, and CLIP-specific defenses like CleanCLIP require on the order of 100k clean images for full fine-tuning. In contrast, TBAR+DECREE uses a much smaller clean subset (and a small forget set) to sanitize an already trained checkpoint.
>
> ### **Scaling coefficient (W3) and theoretical justification (Q3)**
>
> In line with prior work on model editing based on weight interpolations ([1],[2],[3],[4]) we perform a grid search to determine the optimal scaling coefficient for our vectors on a small validation set. This line of work is empirically motivated by linear mode connectivity [5], which shows that, for large neural networks, there often exist low-loss linear paths between related solutions in weight space. However, obtaining formal guarantees on models of this scale and complexity remains an open and challenging problem that falls significantly outside the scope of our work.
>
> Consistent with that literature, our contribution here is empirical: Appendix B.2 of the original submission shows that TBAR is robust to the choice of this parameter, with both ASR and CA varying only mildly around the selected value.
>
> ### **Attack coverage and additional attack evaluations (W4 and Q4)**
> We agree that testing TBAR on a broad set of attacks is important. Beyond the mentioned attacks, the original submission already evaluates SIG (a sinusoidal, feature-like trigger) and BadMerging (a parameter-space stealth attack) in the known-trigger setting, both reported in the appendix. Application to more attacks is an interesting future direction (see also answer to reviewer JcFK on application of TBAR against the Grond attack, a stealthy backdoor in model parameter space for classifiers).

---

> > ### Author Response · Authors · 2025-11-25
> > **Response to reviewer i69k (Part 2 of 2)**
> >
> > ### **Backdoor detection (Q1)**
> > In this work, we do not target the detection problem. TBAR itself is not a detection method: it assumes that either the trigger is known or that we have access to a (possibly approximate) set of triggered examples. Hence, we use DECREE for both its original purpose (detection-only) and extend its usage for backdoor mitigation.
> >
> > ### **Model capacity (Q2)**
> >
> > Our focus in this work is on large pretrained models, where weight disentanglement is primarily induced by the large-scale pretraining. Our experiments on ViT-B vs ViT-L and ConvNeXt/DINO results suggest stronger disentanglement for larger models (consistent with observations in prior literature [2][4], which in turn leads to better CA/ASR tradeoffs. Extending TBAR to very small networks (or limited-capacity networks) is an interesting question. Yet, we don't see scale as a limitation, and we would like to point out that the models and settings we consider are largely used in practice.
> >
> > ### **Missing descriptions (W5)**
> > We respectively disagree. We follow prior work (Carlini and Terzis (2021), Bansal et al. (2023), Liang et al. (2024)) and fully describe the attack pipelines in Section 5.1 (see lines 205-212) and Section 5.3 (see lines 300-305) with further per-attack details outlined in Appendix A.1.
> >
> > ### **Table caption placement (Q5)**
> > As per the conference guidelines, authors should use the iclr2026 template, which explicitly states that "The table number and title always appear before the table."
> >
> > ### **Missing table captions in Appendix (Q6)**
> > We thank the reviewer for pointing this, we have updated the manuscript and included them.
> >
> > [1] Ilharco, Gabriel, et al. Patching open-vocabulary models by interpolating weights. Neurips 2022.
> >
> > [2] Ilharco, Gabriel, et al. Editing models with task arithmetic.  ICLR 2023.
> >
> > [3] Yadav, Prateek, et al. Ties-merging: Resolving interference when merging models. Neurips 2023.
> >
> > [4] Ortiz-Jimenez, et al. Task arithmetic in the tangent space: Improved editing of pre-trained models. Neurips 2023.
> >
> > [5] Frankle, Jonathan, et al. Linear mode connectivity and the lottery ticket hypothesis. ICML 2020.

---

> ### Comment · Reviewer_i69k · 2025-11-26
> **Incremental Contribution on Established Theoretical Grounds**
>
> The authors summarize their core contributions as two-fold: (i) an empirical demonstration that backdoor behaviors are separable from normal functionalities in the weight space, and (ii) a lightweight post-hoc mitigation method leveraging this phenomenon.
>
> However, it must be emphasized that the fundamental theoretical insight—i.e., the separation between backdoor and clean tasks in the weight space—has already been systematically explored and thoroughly established in prior work, notably in "Exploring the Orthogonality and Linearity of Backdoor Attacks". That paper not only provided a rigorous theoretical framework characterizing backdoors through the lenses of orthogonality and linearity, but also comprehensively analyzed how these properties underlie the effectiveness of various defenses.
>
> In this context, the present paper does not introduce significant novelty at the conceptual level. Its main contribution lies in the engineering of a practical mitigation mechanism based on an already well-identified theoretical principle. While the proposed method may offer some practical utility, the core intellectual advancement is limited, as it builds directly upon pre-existing theoretical groundwork without substantially extending it.

---

> ### Author Response · Authors · 2025-11-26
>
> We thank the reviewer for their answer. However, we respectfully but strongly disagree that our core conceptual contribution has already been "thoroughly established" by "Exploring the Orthogonality and Linearity of Backdoor Attacks" [1] by Zhang et al. **None of our main results or conclusions can be directly derived from [1].**
>
> In particular, [1] provides a theoretical characterization of backdoor attacks in a continual learning setup and in terms of gradient *orthogonality* and feature-space linear separability (*linearity*). On the one hand, their theoretical analysis is based on the NTK/linearized regime and explicitly assumes that gradients (or parameter updates) corresponding to clean and backdoored data are orthogonal. Rather, we study how backdoors are encoded in weight space, building on the theory of **weight-disentanglement** (WD) (Ortiz et al., 2024). As discussed in Ortiz et al., WD is **not a unique property of models trained in the NTK regime**, but also applies to standard, non-linearly trained models and it **does not assume orthogonality** of the parameter updates (*task vectors*). Indeed, in our experiments, the clean and trigger task vectors used to measure WD (e.g., Fig. 2) display non-trivial alignment. On the other hand, the *linearity* considered in [1] is defined in the activation space of a small subnetwork (top-3\% "compromised" neurons) and refers to the linear separability of backdoored and clean examples in that space. This is fundamentally different from **linearity of operations in the weight space** in our framework, where simple arithmetic operations on the weights result in corresponding functional behaviors in function space, thanks to WD. In other words, our formalism therefore differs from [1] along several axes: we work with (i) pretrained rather than randomly-initialized models, (ii) standard, non-linearly trained networks rather than models in the NTK regime, (iii) generally non-orthogonal task vectors, (iv) over all model weights rather than a small subset, and (v) linearity in weight space emerging via WD.
>
> Empirically, [1] primarily shows **correlational** evidence between approximate orthogonality and linearity and the success of existing defenses, but they do not provide any concrete removal algorithm based on their analysis. Our work, instead, is **interventional** and **causal** in nature. We show that subtracting the inferred malicious task vectors removes $\approx 99$\% of the backdoor with minimal impact on clean performance. Additionally, our experiments are conducted not on small, supervised classifiers, but on SOTA, contrastively pre-trained vision-language transformer models.
>
> We hope our answer clarifies the novelty of our contributions, which is also highlighted by the reviewer themselves, cf. Strengths 1, and by other reviews as well. To improve and better clarify our placement within the literature, we will add a citation to [1] and include a short discussion of the differences between the frameworks and findings. In light of these clarifications, we kindly ask the reviewer to reconsider their assessment, and we remain available for further discussion.

---

### Official Review · Reviewer_7Znm · 2025-10-31

**Soundness:** 3
**Presentation:** 3
**Contribution:** 2
**Rating:** 4
**Confidence:** 4

**Summary:**

This paper proposes a new backdoor defense, TBAR (Trigger removal by Backdoor ARithmetic), for removing backdoors from large vision-language foundation models such as CLIP. The key insight is that backdoor behaviors are linearly disentangled from benign tasks in the model’s weight space. Leveraging this property, the authors fine-tune the model on a small set of triggered samples to estimate a trigger vector that represents the malicious direction in parameter space, and then subtract it to “unlearn” the backdoor.

**Strengths:**

- The proposal to view backdoor behavior as an independent task direction in the weight space is an inspiring insight that naturally combines model editing with security defense.

- TBAR requires only a small-scale fine-tuning and does not require retraining.

- The presentation of the paper is easy to follow.

**Weaknesses:**

- The number of evaluated backdoor attacks is small. Only BadNet, Blend, and WaNet (BadCLIP in some cases).

- The weight disentanglement demonstration in Figure 2 is not convincing, because it only includes one attack, BadNet.

- The performance drop cannot be ignored in Table 3. In some cases, it exceeds 5%.

- In a trigger-unknown setting, the proposed defense relies on other trigger reverse engineering methods.

**Questions:**

Thanks for the interesting paper. I have a few questions and suggestions.

- The part about known-trigger settings (Section 5) takes the major experiments in the paper. However, it is not practical for the defender to know the trigger. It significantly weakens the contribution of this paper. I suggest that the authors only use the known-trigger setting to analyze the weight disentanglement phenomenon and shorten this section. The experiments should focus more on the setting without explicit knowledge of the attack.

- The section about the weight disentanglement Hypothesis is not necessary. As you provide experimental evidence, you can write it as a phenomenon, and shorten the text to make it more straightforward.

- In the Agnostic attack section, why not try other trigger reverse engineering methods than DECREE?

- Figure 2 is a bit unclear to me. Does it mean that the parts of the model with high $\xi (\alpha_c, \alpha_t)$ values are related to the backdoor? Is it possible to make the backdoor more obvious in another way? For example, Figure 2 in [A] shows that the backdoor-related neurons are prominent than others, or Figure 1 in [B] (although this one is the feature space).

[A] Towards Backdoor Stealthiness in Model Parameter Space

[B] Revisiting the Assumption of Latent Separability for Backdoor Defenses

---

> ### Author Response · Authors · 2025-11-25
> **Response to reviewer 7Znm (Part 1 of 2)**
>
> We thank the reviewer for highlighting several key strengths of our work. We appreciate their recognition
> of the novelty of our approach and they found our presentation easy to follow. In response to their request, we now include t-SNE visualizations showcasing the latent effects of applying TBAR.
>
> ### **Choice of attacks (W1)**
> We agree that evaluating against a diverse set of backdoor attacks is important. In addition to the attacks mentioned by the reviewer, our original submission already included evaluations on two further attacks: (i) *SIG*, a sinusoidal "feature-like" trigger that produces distributed, low-amplitude perturbations rather than a localized patch (Appendix C.1), and (ii) *BadMerging*, a parameter-space stealth attack that explicitly optimizes for a small signature in weight space and robustness under model merging (Appendix B.5). Benchmarking against these attacks is standard in the poisoning literature for CLIP. During the rebuttal, we also showed that TBAR is effective against the Grond attack (please refer to the answer to reviewer JcFK).
>
> ### **Weight disentanglement visualizations (W2)**
>
> As noted in the caption of Figure 2, additional visualizations for other attacks were included in the Appendix. Importantly, these results confirm the picture presented for BadNet, which is thus not limited to this attack.
>
> ### **Additional demonstrations (Q4)**
> Figure 2 visualizes the interaction between two task vectors: a benign task vector and a trigger task vector. The heatmaps report the average prediction disagreement between a composed model (obtained by adding both task vectors to the pretrained weights) and the corresponding single-task edited models, i.e., (pretrained + clean task) on clean inputs and (pretrained + trigger task) on poisoned inputs. Low error indicates that the clean and trigger components behave as approximately independent directions in weight space. This metric is defined in Equation 2 and adopted by many works in the model-merging literature.
> To complement these results, we now also provide t-SNE plots (see Figure 13 in Appendix C.4 of the updated manuscript), showing that after applying TBAR, the feature representations of clean and triggered samples no longer form separate clusters in feature space, indicating successful backdoor removal.

---

> > ### Author Response · Authors · 2025-11-25
> > **Response to reviewer 7Znm (Part 2 of 2)**
> >
> > ### **Performance trade-offs (W3)**
> > In some of the most challenging settings in Table 3, TBAR does incur a small drop in clean accuracy, which reflects an inherent robustness–accuracy trade-off (e.g., "no free lunch" effect (Tsipras et al. (2018) [1])) when fully suppressing a strong backdoor. However, our results show that TBAR offers a significantly more favorable accuracy / ASR trade-off than large-scale clean-data fine-tuning defenses, which is what counts in practice. Moreover, this tradeoff further tends to improve with increasing model scale. We will comment on this in the revised version.
> >
> > ### **Choice of using DECREE (W4 and Q3)**
> >
> > TBAR is compatible with any unsupervised reverse engineering method. The specific choice to use DECREE was driven by CLIP's open-vocabulary nature and the fact that, to the best of our knowledge, DECREE is the best available open-source method. Notice that several alternative trigger inversion techniques (e.g., [2],[3],[4],[5],[6],[7],[8]) rely on a class-induced supervisory signal and are thus incompatible with image-caption data due to their stronger assumptions.
> >
> > ### **Known-trigger setting discussion and results (Q1)**
> >
> > We agree that, from a deployment perspective, the trigger-unknown setting is more realistic. The known-trigger setting in Section 5 is intentionally a controlled environment that lets us isolate and validate the weight-space dynamics of TBAR itself (disentanglement, CA/ASR trade-offs). Once this mechanism is understood and shown to work in the controlled case, we then move to the more challenging trigger-unknown setting in Section 6, where TBAR is combined with DECREE. We therefore view the known-trigger experiments as necessary for scientifically studying how backdoors are encoded in weight space and for characterizing the method rather than as a separate, unrealistic use case. We will clarify this motivation in the revision.
> >
> > ### **Placement of weight disentanglement hypothesis (Q2)**
> > We thank the reviewer for this suggestion. Our intention in introducing Eq. (3) and the accompanying hypothesis was to make explicit the assumption that TBAR relies on, following a standard scientific procedure, i.e., make an assumption and then test it empirically. Are you proposing to remove the explicit "Hypothesis" label and rewrite the surrounding text, while keeping only the minimal formalism needed to explain how TBAR operates?
> >
> >
> >
> > [1] Tsipras, Dimitris, et al. Robustness may be at odds with accuracy. arXiv:1805.12152, 2018.
> >
> > [2] Wang, Bolun, et al. Neural cleanse: Identifying and mitigating backdoor attacks in neural networks. IEEE S&P 2019.
> >
> > [3] Liu, Yingqi, et al. Abs: Scanning neural networks for back-doors by artificial brain stimulation. ACM SIGSAC 2019.
> >
> > [4] Guo, Wenbo, et al. Tabor: A highly accurate approach to inspecting and restoring trojan backdoors in ai systems. arXiv:1908.01763, 2019.
> >
> > [5] Wang, Zhenting, et al. Rethinking the reverse-engineering of trojan triggers. Neurips 2022.
> >
> > [6] Shen, Guangyu, et al. Backdoor scanning for deep neural networks through k-arm optimization. ICML, 2021.
> >
> > [7] Wang, Zhenting, et al. Unicorn: A unified backdoor trigger inversion framework. ICLR 2023.
> >
> > [8] Niu, Yuwei, et al. Test-Time Multimodal Backdoor Detection by Contrastive Prompting. ICML 2025.
> >
> > [B] Revisiting the Assumption of Latent Separability for Backdoor Defenses

---

### Official Review · Reviewer_JuRf · 2025-11-01

**Soundness:** 3
**Presentation:** 4
**Contribution:** 3
**Rating:** 4
**Confidence:** 3

**Summary:**

The paper proposes TBAR, a lightweight backdoor unlearning method for vision-language pretrained models like CLIP. It leverages the observed disentanglement of backdoor and clean knowledge in weight space: by fine-tuning on a small set of triggered samples to estimate a “trigger vector,” the method subtracts this vector to surgically remove the backdoor. Experiments show TBAR achieves high backdoor unlearning performance while preserving ~96% clean accuracy.

**Strengths:**

1. The paper provides compelling evidence that backdoor behaviors in CLIP are linearly disentangled from clean tasks in weight space, a non-trivial finding that enables precise surgical removal.
2. TBAR requires only few triggered samples to achieve near-complete unlearning, using less data than clean-data fine-tuning defenses while yielding better CA/ASR trade-offs.
3. The paper is well-structured, and the experimental evaluation is thorough.

**Weaknesses:**

1. The proposed method should be clearly distinguished from existing approaches such as Anti-Backdoor Learning (ABL) [1], which shares part of the unlearning pipeline. Extensive experimental comparison would help clarify the novelty and relative advantages.
2. How clean-task performance is preserved when directly subtracting model parameters? In related fields such as model merging, naïvely combining parameters from different models often degrades performance. The authors should provide a more detailed justification or analysis for why this operation does not harm utility in their setting.
3. How is α determined in practice? If improperly set, could it severely degrade either clean accuracy or backdoor removal efficacy?
4. In Step 3 of Figure 1, the method relies on “similarly constructed triggered data” to estimate the backdoor direction τₜ. However, it is unclear how such triggered samples are obtained in practice—especially in a realistic threat model where the attacker’s trigger may be unknown. Clarifying the accessibility and construction of this data is essential, as it directly impacts the feasibility and applicability of the proposed approach.

[1] Li Y, Lyu X, Koren N, et al. Anti-backdoor learning: Training clean models on poisoned data[J]. Advances in Neural Information Processing Systems, 2021, 34: 14900-14912.

**Questions:**

Address the weakness  above

---

> ### Author Response · Authors · 2025-11-25
> **Response to reviewer JuRf**
>
> We thank the reviewer for their time and for recognizing several strengths of our work,
> including the perspective of weight-space disentanglement, the efficiency of TBAR, and its strong empirical performance. Below, we address their concerns.
>
> ### **Comparison to Anti-Backdoor Learning (W1)**
> We agree that Anti-Backdoor Learning (ABL) [Li et al., 2021] is a strong training-time defense, but it operates in a different regime: the goal of ABL, is to train a model on a mix of clean and poisoned data, whereas TBAR is designed to sanitize an already-backdoored CLIP model via weight-space editing. Conceptually, ABL consists of two stages: detecting poisoned samples in the pre-training data and then applying a loss-maximization (gradient-ascent) objective on those samples. Where applicable, we already include a gradient-ascent–based unlearning baseline that mirrors this part of ABL’s pipeline. In addition, CleanCLIP, directly adapted ABL to multimodal contrastive pretraining and reports that it is *not effective* at reducing ASR across several backdoor attacks.
>
> ### **Clean accuracy and ASR removal trade-offs (W2 and W3)**
>
> We thank the reviewer for raising this question. Arguably, our setting is simpler than traditional model merging in several aspects. First, model merging typically combines *multiple* task vectors to obtain a multitask model that retains several distinct skills simultaneously. This naturally introduces interference between tasks (e.g., arising from redundant parameters or sign disagreements across task vectors), which methods such as TIES are explicitly designed to mitigate. In contrast, TBAR edits the model along a *single* direction associated with the backdoor behavior, rather than averaging several task-specific models.
> Second, we do not apply a fixed, "large" subtraction. Instead, we select the scaling coefficient $\alpha$ via grid search on a small validation set, explicitly balancing backdoor removal and clean-task performance. Depending on the setting (known vs unknown trigger), the validation criterion combines clean and triggered samples, and we enforce a clean-accuracy threshold below which we do not further increase the magnitude of the update. The final CA/ASR numbers reported in the tables are then measured on a disjoint test set. Empirically, this procedure yields small drops in clean accuracy while almost completely eliminating the backdoor, as also illustrated by the sensitivity analysis in Appendix B.2.
>
> ### **Figure caption (W4)**
> Figure 1 was meant for illustration purposes to simulate both trigger-known and trigger-unknown settings. In order to avoid confusion, we changed "similarly constructed triggered data" to "reverse engineered triggered data" in the figure caption. Finally, we want to point out that we mentioned reverse-engineered triggers in both the abstract and the introduction, and all details regarding their construction are already explained in Section 6.

---

### Official Review · Reviewer_JcFK · 2025-11-09

**Soundness:** 2
**Presentation:** 3
**Contribution:** 2
**Rating:** 2
**Confidence:** 4

**Summary:**

This paper investigates methods for removing backdoors in the model parameter space of vision foundation models. The authors claim that benign and backdoor behaviour are reflected in the model weights, where different components are linearly separable. With this intuition, the authors propose a parameter-space backdoor removal method, Trigger removal by Backdoor ARithmetic (TBAR). TBAR operates in the parameter space by conducting task negation, which is inspired by and based on the model edit. Several experiments on different datasets show the effectiveness of the TBAR.

**Strengths:**

This paper is well presented. The idea of backdoor defense by model editing is promising.

**Weaknesses:**

Backdoor baselines are weak. The paper primarily compares TBAR with basic fine–tuning–based unlearning and several standard backdoor defense methods. These baselines are relatively weak compared with the state-of-the-art backdoor defenses, such as [1]. Without including stronger and more diverse baselines, it is difficult to assess how much of TBAR’s advantage comes from its intrinsic effectiveness.

Adaptive backdoor attacks are not considered. The evaluation does not include adaptive or defense-aware attackers who might deliberately design entangled or non-linear backdoor patterns to resist task-vector subtraction. Since TBAR relies on the assumption that the backdoor and clean tasks are linearly separable in weight space, a knowledgeable adversary could craft more integrated triggers that invalidate this assumption. The absence of such adaptive attack scenarios leaves uncertainty about TBAR’s robustness in real-world adversarial settings, where attackers can adapt to the defense strategy. At least, TBAR needs to take stealthy parameter space backdoor attacks into account, such as [b].


The connection with machine unlearning is good, but somewhat far-fetched. The paper positions TBAR as a form of machine unlearning, but the connection is mainly conceptual. Traditional machine unlearning focuses on data-level forgetting with formal guarantees of data removal. In contrast, TBAR performs parameter-space editing by subtracting a task vector without guarantees.

[a] Towards Reliable and Efficient Backdoor Trigger Inversion via Decoupling Benign Features. ICLR 2024.

[b] Towards Backdoor Stealthiness in Model Parameter Space. CCS 2025.

**Questions:**

Please discuss the potential of TBAR against stronger backdoors, including adaptive backdoor attacks.
Please discuss other stronger baselines.
Please adjust and articulate the connection with machine learning.

---

> ### Author Response · Authors · 2025-11-25
> **Response to reviewer JcFK (Part 1 of 2)**
>
> We thank the reviewer for their feedback and for highlighting the clarity and novelty of our work. We address their comments below.
>
> ### **Choice of baselines (W1 and Q2)**
>
> We respectfully disagree that our backdoor baselines are weak. In the post-hoc, multimodal CLIP setting, CleanCLIP, RoCLIP, and Contrastive FT with a 100k order of data budget are widely used and considered strong state-of-the-art baselines [1][2][3]. This setting is fundamentally different from [a], which targets small closed-set classifiers which are backdoored from scratch. Whereas we focus on practically relevant, large-scale CLIP models backdoored after pretraining/finetuning, with the goal of removing the backdoor while preserving clean accuracy. In fact, notice that retraining such models from scratch is not an option given the scale of pretraining and thereof.
>
> Nonetheless, to directly address the reviewer’s concern, we also adapt BTI-DBF(U) [a] to CLIP by training a U-Net trigger generator on the backdoored model and applying their algorithm with CLIP-based losses under a similar 1.5k data budget as TBAR. In contrast to our method, this adapted defense is ineffective against Blended and BadCLIP and can suffer a larger clean-accuracy drop under attacks such as BadNet.
>
> |                                 | **BadNet**       |                     |     | **Blended**      |                     |     | **WaNet**        |                     |     | **BadCLIP**      |                     |
> | :------------------------------ | :--------------: | :-----------------: | :-- | :--------------: | :-----------------: | :-- | :--------------: | :-----------------: | :-- | :--------------: | :-----------------: |
> |                                 | CA  | ASR  |     | CA | ASR  |     | CA  | ASR  |     | CA  | ASR  |
> | Zero-Shot                       | 63\.34%          | 00\.00%             |     | 63\.34%          | 00\.00%             |     | 63\.34%          | 00\.00%             |     | 63\.34%          | 00\.00%             |
> | Backdoored                      | 61\.69%          | 84\.48%             |     | 61\.39%          | 99\.67%             |     | 61\.32%          | 93\.12%             |     | 61\.41%          | 99\.98%             |
> | *reverse-engineered unlearning* |                  |                     |     |                  |                     |     |                  |                     |     |                  |                     |
> | GA+DECREE                       | 60\.41%          | 08\.30%             |     | 56\.92%          | 76\.40%             |     | 60\.22%          | 35\.67%             |     | N/A              | N/A                 |
> | TBAR+DECREE                     | 60\.29%          | 00\.33%             |     | 55\.56%          | 00\.90%             |     | 56\.85%          | 00\.64%             |     | N/A              | N/A                 |
> | BTI-DBF (U)                     | 56\.92%          | 00\.32%             |     | 57\.67%          | 44\.55%             |     | 57\.68%          | 00\.02%             |     | 56\.94%          | 99\.98%             |
>
> ### **Choice of attacks (W2 and Q1)**
> We respectfully disagree that adaptive attacks are not considered in our evaluations at all. In the submission, we already include BadCLIP (a CLIP-specific attack that is designed to remain effective under fine-tuning defenses) and BadMerging (a parameter-space stealth attack that explicitly optimizes backdoor robustness under model merging).
>
> In addition, following the reviewer’s suggestion, we have now implemented the Grond attack from [b] on CIFAR-10, using both ConvNeXt-base and ViT-B/32 classifiers. We test both architectures using UPGD noise derived from the clean ConvNext-base model to measure cross-architecture attack robustness. In these new experiments, TBAR reduces Grond’s ASR to less than 2\% with only 2\% clean-accuracy loss, showing that the method remains effective even against a backdoor that is explicitly optimized for parameter-space stealthiness.
>
> | **Arch**          | **init CA** | **init ASR** | **TBAR CA** | **TBAR ASR** |
> | :---------------- | :---------: | :----------: | :---------: | :----------: |
> | **ConvNext-base** | 97\.73      | 99\.84       | 95\.53      | 1\.52        |
> | **ViT-B/32**      | 97\.65      | 100          | 96\.00      | 0\.60        |
>
> We agree that a fully defense-aware adversary could, in principle, try to deliberately entangle backdoor and clean behavior to violate our linear-decomposition assumption. However, we would expect such an attack to significantly alter the model's (clean) base knowledge. Studying whether such attacks can be carried out without compromising the clean behavior of foundation models is an interesting direction for future work.

---

> > ### Author Response · Authors · 2025-11-25
> > **Response to reviewer JcFK (Part 2 of 2)**
> >
> > ### **Connection to machine unlearning (W3 and Q3)**
> > As noted in our related work section (see L456-458) there exists two type of machine unlearning works: exact unlearning and approximate unlearning. Our intention was not to claim that TBAR provides certified unlearning in the sense of Cao \& Yang, (2015); Bourtoule et al. (2021), but rather to show that weight-space task arithmetic offers a practical mechanism for targeted removal of specific behaviors (backdoor tasks) in large pre-trained encoders, similarly to standard approximate unlearning methods, which struggle with data poisoning (see, e.g., Pawelczyk et al. (2024) in the case of small-scale models).
> >
> > In the revision, we will explicitly rephrase the positioning to stress that TBAR is a model-editing-based backdoor removal method motivated by approximate unlearning, to avoid suggesting that it satisfies the formal guarantees studied in the exact unlearning literature and that are hard to obtain at this scale (large transformer vision-language models trained on ImageNet or captioning tasks as CC3M).
> >
> > [1] Bansal, Hritik, et al. Cleanclip: Mitigating data poisoning attacks in multimodal contrastive learning. ICCV 2023.
> >
> > [2] Yang, Wenhan, Jingdong Gao, and Baharan Mirzasoleiman. Robust contrastive language-image pretraining against data poisoning and backdoor attacks. Neurips 2023.
> >
> > [3] Yang, Wenhan, Jingdong Gao, and Baharan Mirzasoleiman. Better safe than sorry: pre-training CLIP against targeted data poisoning and backdoor attacks. ICLR 2024.
> >
> > [a] Towards Reliable and Efficient Backdoor Trigger Inversion via Decoupling Benign Features. ICLR 2024.
> >
> > [b] Towards Backdoor Stealthiness in Model Parameter Space. CCS 2025.

---

### Meta-Review · Area_Chair_ciiM · 2025-12-31

**Summary:**

This work proposed a backdoor defense for CLIP from the machine unlearning perspective, based on the observation that backdoors are disentangled from other benign tasks.

The major concerns from four reviewers and my thoughts include:
1. The conceptual novelty of the disentanglement of backdoor and benign tasks. I agree with that this phenomenon has been discussed by lots of existing works explicitly or implicitly. This work provides one additional analysis about it, but without theoretical support. This work didn't push forward the conceptual boundary of backdoor.
2. The limitation of trigger-discovery-based defenses. The proposed method depends on the trigger reengineering, which is still a challenging task, especially for adaptive, sample-specific triggers. The method didn't verify its effectiveness under these complex triggers.  If the trigger is well recovered, lots of defenses could achieve very good performance.
3. Limited baselines and attacks. Since the proposed method is empirically verified, without theoretical support, more baselines and attacks (even considering the additional ones in rebuttal) are expected.

Above intrinsic points are not (and cannot be) addressed in the rebuttal.

Thus, the recommendation is reject.

**Reviewer Concerns:**

please refer to the summary.

**Reviewer Scores:**

please refer to the summary.

---

### Decision · Program_Chairs · 2026-01-26

Reject